# Bariatric surgery induces a new gastric mucosa phenotype with increased functional glucagon-like peptide-1 expressing cells

Lara Ribeiro-Parenti[1,2,10], Anne-Charlotte Jarry[1,10], Jean-Baptiste Cavin[1], Alexandra Willemetz[1], Johanne Le Beyec[1,3], Aurélie Sannier[4], Samira Benadda[1,5], Anne-Laure Pelletier[1], Muriel Hourseau[4], Thibaut Léger [6,7], Bastien Morlet[6], Anne Couvelard[1,4], Younes Anini[8], Simon Msika[1,2], Jean-Pierre Marmuse[1,2], Sévérine Ledoux[1,9], Maude Le Gall [1✉] & André Bado [1✉]

Glucagon-Like Peptide-1 (GLP-1) undergoes rapid inactivation by dipeptidyl peptidase-4 (DPP4) suggesting that target receptors may be activated by locally produced GLP-1. Here we describe GLP-1 positive cells in the rat and human stomach and found these cells co-expressing ghrelin or somatostatin and able to secrete active GLP-1 in the rats. In lean rats, a gastric load of glucose induces a rapid and parallel rise in GLP-1 levels in both the gastric and the portal veins. This rise in portal GLP-1 levels was abrogated in HFD obese rats but restored after vertical sleeve gastrectomy (VSG) surgery. Finally, obese rats and individuals operated on Roux-en-Y gastric bypass and SG display a new gastric mucosa phenotype with hyperplasia of the mucus neck cells concomitant with increased density of GLP-1 positive cells. This report brings to light the contribution of gastric GLP-1 expressing cells that undergo plasticity changes after bariatric surgeries, to circulating GLP-1 levels.

[1] Université de Paris, Inserm U1149, Centre de Recherche sur l'inflammation, Paris, France. [2] Service de Chirurgie Générale Œsogastrique et Bariatrique, Hôpital Bichat - Claude-Bernard, Assistance Publique-Hôpitaux de Paris, Paris, France. [3] Sorbonne Université, AP-HP, Hôpital Pitié-Salpêtrière-Charles Foix, Biochimie Endocrinienne et Oncologique, Paris, France. [4] Department of Pathology Bichat Hospital, AP-HP, 75018 Paris, France. [5] Cell and Tissue Imaging Platform, Inserm, U1149, CNRS, ERL8252, 75018 Paris, France. [6] Université de Paris, Mass Spectrometry Laboratory, Institut Jacques Monod, UMR 7592, CNRS, 75205 Paris, France. [7] Université Rennes, Inserm, EHESP, Irset (Institut de recherche en santé, environnement et travail) - UMR_S 1085, 35000 Rennes, France. [8] Department of Obstetrics and Gynecology, Dalhousie University, IWK Health Centre, Halifax, New Brunswick, Canada. [9] Service des Explorations Fonctionnelles Hôpital Louis Mourier, AP-HP, Centre Intégré Nord Francilien de prise en charge de l'Obésité (CINFO), 92701 Colombes, France. [10] These authors contributed equally: Lara Ribeiro-Parenti, Anne-Charlotte Jarry. ✉email: maude.le-gall@inserm.fr; andre.bado@inserm.fr

Glucagon-like peptide-1 (GLP-1) is a gastrointestinal peptide, produced and secreted by endocrine proglucagon-producing L cells scattered along the gastrointestinal tract and most abundant in the ileum and colon. In the intestine, GLP-1 is produced upon prohormone convertase 1/3 (PC1/3)-mediated processing of proglucagon, which also produces glicentin, oxyntomodulin, GLP-2, and intervening peptide 2[1,2]. One of the most important function of GLP-1 is to act as an incretin by stimulating glucose-dependent release of insulin from pancreatic β-cells[3–5] while suppressing glucagon secretion by α-cells[6], in response to food ingestion. However, the possibility that intestinal-produced GLP-1 is secreted into the circulation at concentrations sufficient to activate its receptors on distant target tissues remains a matter of debate as intestinally secreted GLP-1 is rapidly degraded by dipeptidyl peptidase-4 (DPP-4). Recently, it was demonstrated that bioactive GLP-1 is produced in the islet α-cell and that production is increased in response to inflammation[7,8]. Moreover, using mouse models of organ-specific invalidation of GLP-1 production, it was suggested that the pancreatic but not the intestinal source of GLP-1 was necessary for glucose regulation[9].

Among its extra-pancreatic effects, GLP-1 decreases gastric acid secretion directly by inhibition of parietal cells and indirectly by stimulation of somatostatin (SST) secretion by D cells[10–12]. GLP-1 also reduces gastric emptying[13]. During a meal, this effect reduces the intestinal delivery of gastric contents resulting in a smaller glucose excursion, one of the mechanisms of action for diabetes therapeutic drugs[14]. The characterization of GLP-1 receptors on parietal cells, gastric endocrine cells, and myenteric and the vagal afferent neurons makes the stomach a key target for GLP-1[10,11,15]. Given the rapid degradation of intestinally secreted GLP-1 by DPP-4, there are suggestions that these GLP-1 receptors are activated locally—i.e., by gastric-produced GLP-1.

Here we show that GLP-1-expressing cells are located in the stomach and contribute to the active GLP-1 levels in the portal vein. These cells undergo plasticity changes after bariatric surgeries in obese rats and individuals and may locally modulate gastric functions contributing to the control of glucose homeostasis.

## Results

**GLP-1-positive cells are present in the gastric mucosa.** We first detected GLP-1 immunoreactive cells in the corpus/fundus region in both the rat and human stomach (Fig. 1a, b). The presence of these GLP-1-positive cells was consistent in rats with the mucosal expression of the key enzyme for proglucagon processing into GLP-1 (i.e., *Pcsk1/3* coding for PC1/3) and of *Pax6*, required for L cell differentiation (Supplementary Fig. 1). Moreover, *Gcg* mRNA together with *Peptide YY* (*Pyy*) mRNA were readily detected from the stomach to large intestine, with higher mRNA levels found in the distal intestine. Levels of *Gcg*, *Pyy*, *Pcsk1/3*, and *Pax6* mRNAs in the gastric mucosa and intestine did not significantly change after 4 months of high-fat diet (HFD) in rats (Supplementary Fig. 1).

Using a specific total GLP-1 radioimmune assay (RIA; Supplementary Fig. 2), GLP-1 protein was detected in acid-ethanol extracts of rat and human gastric mucosa. In the rat stomach, mucosal GLP-1 protein levels were $50 \pm 11.2$ pmoles and $25 \pm 6$ pmoles/g wet mucosa of fundus and antrum, respectively, compared to $243 \pm 12.5$ pmoles/g wet colon mucosa (Fig. 1a). GLP-1 protein was also readily quantifiable from gastric biopsies of obese individuals with higher levels of this peptide in the fundus compared to the antrum (Fig. 1b).

Further characterization by mass spectrometry (MS) (Fig. 1c) of acid-ethanol extracts of rat gastric and ileal mucosa identified proglucagon, GLP-1, and GLP-2 fragments consistent with PC1/3 cleavage sites after trypsin, Lys-C, chymotrypsin, and Glu-C

digestion. In the gastric mucosa, four peptides (FINWLIQTKITD with Glu-C and EFIAWLVK, DFPEEVAIAEELGR, DFINWLIQTK with trypsin) belonging to GLP-1 and GLP-2 sequences were identified. All peptides obtained by trypsin, Glu-C, Lys-C, and chymotrypsin digestion and the quantitative proteomics experiments from the trypsin data set are shown in Supplementary Table 1 and the corresponding MS spectrum of the peptides fragments are shown in Supplementary Fig. 3.

We next evaluated whether the GLP-1-positive cells also contain hormones from other gastric endocrine cells. As shown in Fig. 1d, confocal analyses revealed that GLP-1-positive cells are also immunoreactive for ghrelin in the fundus and SST in the antrum of the stomach.

We next question whether this stomach GLP-1 was active comparing total and active GLP-1 content in the stomach with distal regions of the gut, i.e., ileum and colon. As shown in Fig. 1e, ileum and colon contain threefold and sixfold, respectively, higher amount of total GLP-1 compared to stomach. Moreover, in the ileum and colon, active GLP-1 contents were actually 2- and 1.5-fold, respectively, higher than the stomach active GLP-1 content (Fig. 1f).

**Gastric GLP-1-positive cells are responsive to stimulation ex vivo and in vivo.** Fundus, antrum, or colon fragments from normal diet fed (ND) lean rat led to spontaneous release of GLP-1 ex vivo (Fig. 2a). In the stomach, this basal GLP-1 release was 7.4 pmoles/mg protein in the fundus and 9.14 pmoles/mg protein in the antrum (Fig. 2a). This basal GLP-1 release was 15 times higher in the colon than in the stomach ($P < 0.01$). As shown in Fig. 2b, incubation of fundic fragments with the phosphodiesterase inhibitor IBMX resulted in a +70% increase in GLP-1 release ($P < 0.001$ vs. without IBMX). Similarly, a +80% increase in GLP-1 release was observed for antral fragments ($P < 0.01$ vs. without IBMX). Incubation of colon fragments with IBMX resulted in greater increase in GLP-1 release (+100%, $P < 0.001$ vs. without IBMX) than that observed with gastric fragments. In HFD obese rat stomach, ex vivo incubation of the fundus fragments with IBMX (Supplementary Fig. 4) still significantly increased the GLP-1 release, but in the antrum, this GLP-1 release was not significantly reduced, suggesting an overall reduction of gastric GLP-1 release in response to obesity condition. Thus the stomach contains functional cells able to produce and secrete GLP-1 ex vivo in response to IBMX, an effect that is impaired in the antrum of obese rat.

Subsequently, we examined the contribution of this gastric pool of GLP-1 to circulating GLP-1 levels in vivo. To this end, glucose was directly delivered into the stomach or the duodenum of anesthetized rats with a pylorus ligature in order to limit glucose exposure to the intestine or stomach, respectively. Blood was sampled from the portal vein or the gastric vein (Fig. 2c) for total or active GLP-1 assay before and 15 or 30 min after glucose delivery.

As shown in Fig. 2d vs. Fig. 2f, in the basal state, the total GLP-1 levels in the portal vein were significantly 2.7-fold higher ($P < 0.01$) than total GLP-1 levels in the gastric vein. An intragastric load of glucose (in a volume that did not induce any distension of the stomach) led to a significant though transient increase of total GLP-1 levels in the portal vein after 15 min that return to basal after 30 min (Fig. 2d). On the other hand, bypassing the stomach and directly delivering glucose into the duodenum resulted in a rapid rise of total GLP-1 levels in the portal vein after 15 min (+194 pM vs. phosphate-buffered saline (PBS)) that persisted over 30 min (Fig. 2d).

The rise of total GLP-1 levels in the portal vein in response to the intragastric load of glucose was paralleled with a rise of total GLP-1 levels in the gastric vein (Fig. 2e). Moreover, the

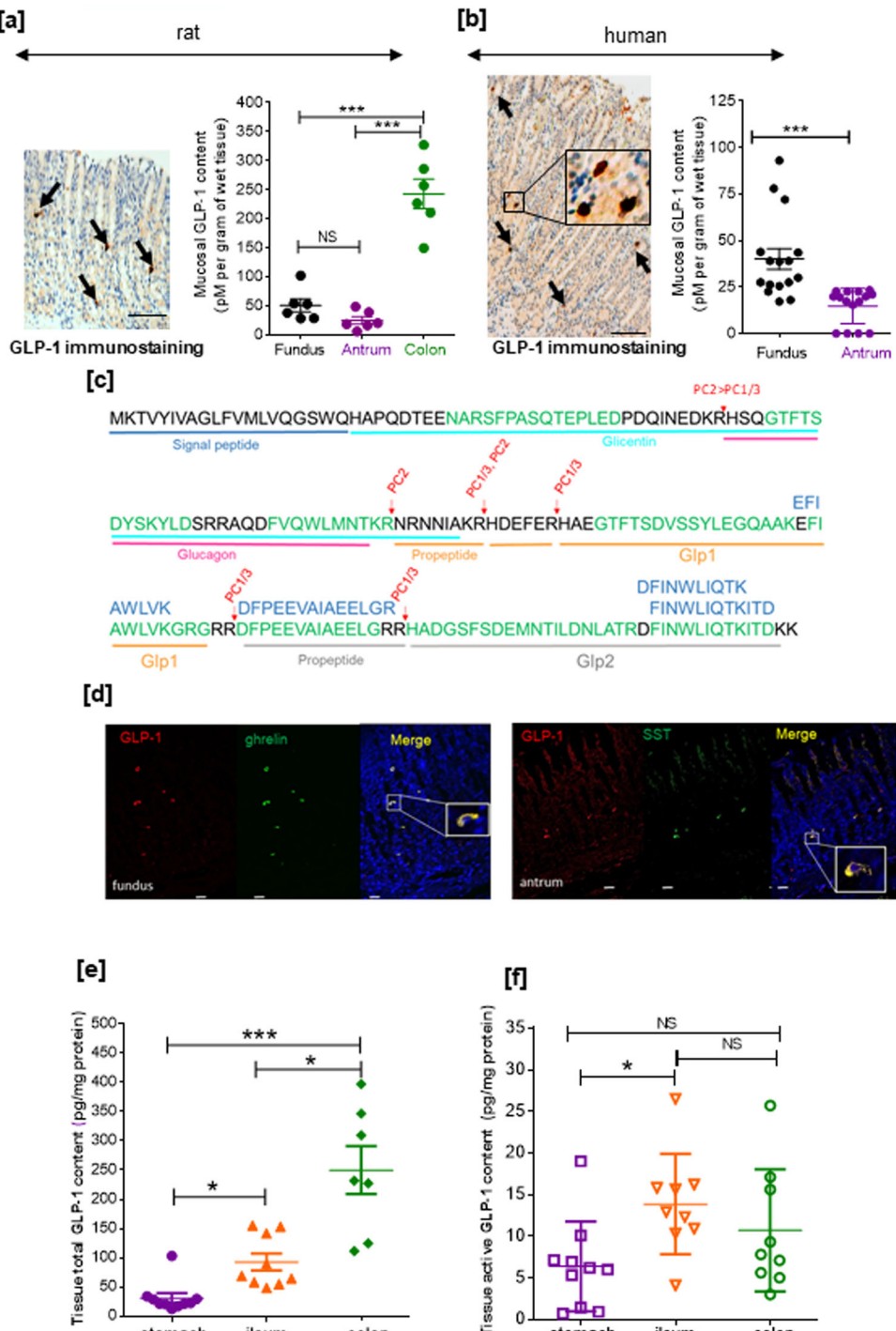

increment of total GLP-1 levels after 15 min was +133 pM (vs. PBS) in the portal vein and +140 pM (vs. PBS) in the gastric vein suggesting a contribution of the GLP-1 detected in the gastric vein to the GLP-1 detected in the portal vein. Finally, the rise of total GLP-1 levels in the gastric vein was accompanied by a rise of active GLP-1 levels (Fig. 2f) indicating that part of the GLP-1 released by the stomach in response to a glucose load is active.

**Obese individuals, operated on bariatric surgery, exhibit a remodeled fundic mucosa with increased densities of GLP-1 cells.** Histologically analyses of fundic mucosa from obese

subjects and obese subjects operated on vertical sleeve gastrectomy (VSG) or Roux-en-Y gastric bypass (RYGB) revealed deep morphological changes and remodeling of the remaining fundic mucosa (Fig. 3a). First, we observed an increase in the gastric pit height associated with an expansion and hyperplasia of the mucous neck cells that may reflect foveolar hypertrophy. Second, an increased number of Periodic Acid Schiff (PAS)-positive cells (i.e., neutral mucin-expressing cells) in the neck region of the glands was observed. Third, there was a significant increase in Ki-67 proliferating cells in the fundic glands after RYGB (sixfold) and VSG (threefold) compared to obese subjects (Fig. 3c). This remodeling of fundic mucosa was concomitant to changes in the density of GLP-1-positive cells. Indeed, in the

**Fig. 1 Expression of GLP-1 in rat and human stomach cells.** GLP-1-positive cells and mucosal GLP-1 content in the rat (**a**) and human (**b**) stomach. Representative immunostaining of GLP-1 in fundic mucosa sections from an HFD rat and an obese individual. Bar scale 100 μm. Arrows indicate GLP-1-positive cells. Insert: a high magnification of human fundic GLP-1-positive cells. Quantification of mucosal GLP-1 content from HFD obese rat stomach and colon mucosal scrapings ($n = 6$ rats) and from gastric biopsies ($n = 15$ for fundus and $n = 16$ for antrum) of obese candidates for bariatric surgery. Results are presented as scatter data plots with mean ± SEM of mucosal GLP-1 protein content. Each point corresponded to one obese subject. One-way ANOVA followed by Tukey's multiple comparison test were used to analyze data in the rat (**a**), and Mann–Whitney test was used compare data in human (**b**). ***$P < 0.001$; NS not significant. **c** LC-MS/MS identification of GLP-1. Principal peptides identified in acid-ethanol gastric mucosa extracts from rat ileum and stomach are represented in green and blue, respectively. Red arrows show physiological cleavage sites by PC1/3. Identified peptides are further described in Supplementary Table 1. **d** Immunoreactivity of GLP-1, ghrelin, and somatostatin (SST) in stomach. Confocal analysis and representative photomicrographs of GLP-1 (red), ghrelin (green), and SST (green) positive cells in rat gastric mucosa. Nuclei were stained with Hoechst solution (blue). Bar Scale 20 μm. Note that the GLP-1-positive cells are also positive for ghrelin in the fundic mucosa and somatostatin in the antrum. Inserts: high magnification of these cells. Total (**e**) and active (**f**) GLP-1 contents in the rat stomach, ileum, and colon. Quantification of total and active GLP-1 content from lean rat stomach, ileum, and colon mucosal scrapings. Results are expressed as GLP-1 in pg/mg protein and presented as scatter data plots with mean ± SEM. Each point corresponded one rat with $n = 10$ for stomach and $n = 9$ for ileum and colon. Data were analyzed by a one-way ANOVA followed Tukey multiple comparison test *$P < 0.05$; ***$P < 0.001$; NS for not significant. Source data are provided as a Source data file.

small number of patients analyzed, GLP-1-positive cells were present in the corpus/fundus mucosa (Fig. 3b) and their quantification showed a significant increase in the density of GLP-1 cells in the gastric pouch after RYGB but no significant change in the residual VSG fundic mucosa compared to obese subjects (Fig. 3d).

In addition to the increase in density of GLP-1-positive cells in the gastric pouch, the density of ghrelin-positive cells was increased by 2-fold ($P < 0.01$ vs. obese subjects), whereas the density of SST- and 5HT-positive cells did not change significantly (Supplementary Fig. 5). In the remaining fundic mucosa of VSG subjects, the density of ghrelin-, SST-, and 5HT-positive cells was decreased by 2-fold ($P < 0.05$ vs. obese subjects). These data indicate differences in cell plasticity between the two surgical procedures.

In order to characterize the functional relevance of the gastric GLP-1 cells in response to weight loss associated with bariatric surgeries seen in humans, we used our previously validated rat models of VSG and RYGB surgery[16,17] (Supplementary Fig. 6 and Supplementary Text 1). These preclinical models of surgery in HFD-induced obese rats allow access to the stomach and to the small and large intestine for additional analyses that are difficult in obese humans.

**Density of GLP-1 cells increased in HFD obese rats operated on VSG or RYGB.** We previously reported an increase in the total number of GLP-1 cells after RYGB as a direct result of overgrowth of the jejunum mucosa, whereas the density of GLP-1 cells did not significantly change[17–19]. Accordingly, in the ileum and the colon of obese rats operated on RYGB, significant morphological changes occurred (Supplementary Fig. 7). These changes include: (i) an increase in the villi height in the ileum accompanied by an increased number of Ki67 proliferative cells in the crypts and (ii) a significant increase in the depth of the colon crypts and a trend toward an increased number of Ki67 proliferative colonic cells. These morphological changes occurred despite no significant change in the density of GLP-1 cells in the ileum and colon (Supplementary Fig. 7). In the distal intestine (ileum and colon) of obese rats operated on VSG, no significant morphological changes were observed (Supplementary Fig. 7) and no change in the abundance of GLP-1 cells were recorded (Supplementary Fig. 7).

In obese rats that underwent RYGB, the density of GLP-1 cells was increased 2-fold ($P < 0.05$ vs. sham) in the gastric pouch mucosa (Fig. 4). In obese rats that underwent VSG surgery, no significant change in the density of GLP-1 cells occurred in the remaining fundic mucosa (Fig. 5a), consistent with data in human fundic mucosa (Fig. 3d). However, in the antral mucosa of obese rats operated on VSG, the density of GLP-1 cells significantly increased ($+50\%$; $P < 0.05$ vs. sham) (Fig. 5b) and the antrum was macroscopically enlarged (Fig. 5c). Finally, the rise of GLP-1 levels in the portal vein in response to a gastric load of glucose observed in anesthetized lean rats (Fig. 2d), was abrogated in the HFD obese rats (Fig. 5d). Remarkably, after VSG this transient rise of GLP-1 levels in the portal vein upon gastric glucose stimulation was restored (Fig. 5d).

**Discussion**

In this report, we demonstrate the presence of GLP-1-expressing cells in the gastric mucosa of rats and humans and show that gastric mucosa can release detectable GLP-1 into both the gastric and the portal vein 15 min after an oral load of glucose in lean animals. These findings build on data from an earlier report showing that, after distal bowel (ileum and colon) resection, an oral load of glucose still resulted in increased circulating GLP-1 levels, suggesting that GLP-1-producing cells in the upper gut are contributing to these levels[20]. They are also consistent with earlier studies describing a small number of GLP-1-positive cells in the corpus/fundus region of human, dog, pig, and rat stomach[21–24]. Still, the reported data concerning the expression of proglucagon-processed products (i.e., GLP-1, GLP-2, glucagon, etc.) within the stomach are conflictual[25–30]. Our current findings comfort previous observations[22–24,26] but are challenged by recent peptidomic profiling from human stomach[27] although negative reports may never be definitive.

Here we identified by liquid chromatography (LC)-MS-based proteomics proglucagon, GLP-1, and GLP-2 fragments consistent with PC1/3 cleavage sites. In addition, the evidenced GLP-1-positive cells are responsive to stimulation by glucose in vivo, probably through a mechanism involving cAMP and protein kinase A stimulation leading to granule secretion as IBMX, a phosphodiesterase inhibitor known to raise intracellular cAMP, mimicked this effect ex vivo. Noteworthy, in vivo, the presence of glucose into the duodenum induced a long-lasting increase of GLP-1 levels in the portal vein, whereas the increase was transient in both the portal and gastric veins in response to solely gastric glucose stimulation. The kinetics and increment of the transient stimulation were similar in the portal vein and the gastric vein, suggesting the contribution of the gastric-derived GLP-1 secretion to the portal vein GLP-1 levels. Noteworthy, active GLP-1 was readily detected and secreted in the gastric vein indicating that the stomach contains bioactive GLP-1. Conversely, the long-lasting GLP-1 release upon duodenal glucose strongly suggests that stimulation of distal L cells in the jejunum, ileum, and colon by the traveling—although progressively absorbed—glucose is responsible for the persistence of GLP-1 secretion.

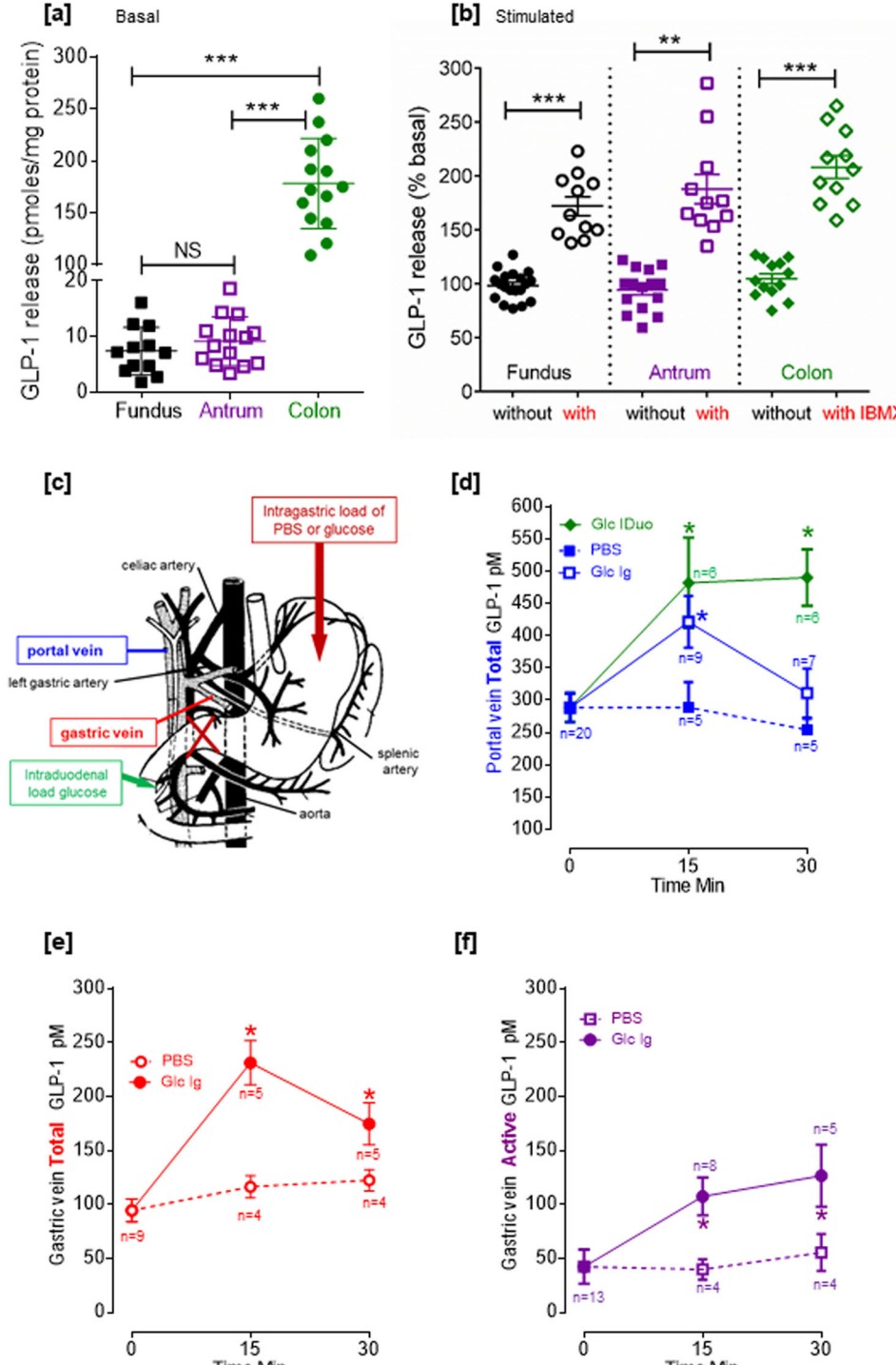

**Fig. 2 The stomach produced active GLP-1 in response to a gastric glucose load.** Spontaneous (**a**) and IBMX-stimulated (**b**) GLP-1 release from fundus, antrum, and colon fragments ex vivo. Stomach and colon from overnight fasted lean rat were collected and incubated for 1 h with PBS (without IBMX) and with 100 μM IBMX. GLP-1 was assayed in the supernatant. Results are expressed as mean ± SEM in pmoles/mg protein for basal release in **a** with $n = 12$ for fundus and $n = 14$ for antrum and the colon and statistically analyzed with one-way ANOVA followed by Tukey's multiple comparison test. ***$P < 0.001$, **$P < 0.01$, and NS not significant. When stimulated with IBMX, results are expressed as mean ± SEM in percentage of basal release (without IBMX $n = 19$ for fundus, $n = 15$ for antrum, and $n = 13$ antrum and with IBMX $n = 11$ for fundus, antrum, and colon) and statistical analysis was performed using Mann–Whitney test to compared with and without IBMX **$P < 0.01$, ***$P < 0.001$. **c** Scheme of the stomach showing the portal and gastric vein blood sampling and loading of PBS and glucose. **d** In vivo intragastric load of glucose increases total GLP-1 levels in the portal vein of anesthetized rat. Total GLP-1 levels after intragastric (Ig) or intraduodenal (IDuo) load of glucose (2 g/kg BW) or PBS measured in the portal vein (blue: Ig Glc and PBS; green: IDuo Glc). Data are the mean ± SEM of $n$ values indicated at each time point on the graph. Data were analyzed by a two-way ANOVA. *$P < 0.05$ vs. PBS. Total (**e**) and active (**f**) GLP-1 levels increase after intragastric load of glucose in the gastric vein of anesthetized rat. Total and active GLP-1 levels after an intragastric (Ig) load of glucose (2 g/kg BW) or PBS measured in the gastric vein. Data are the mean ± SEM, $n$ values indicated at each time point on the graph, and data were analyzed by a two-way ANOVA. *$P < 0.05$ vs. PBS. Source data are provided as a Source data file.

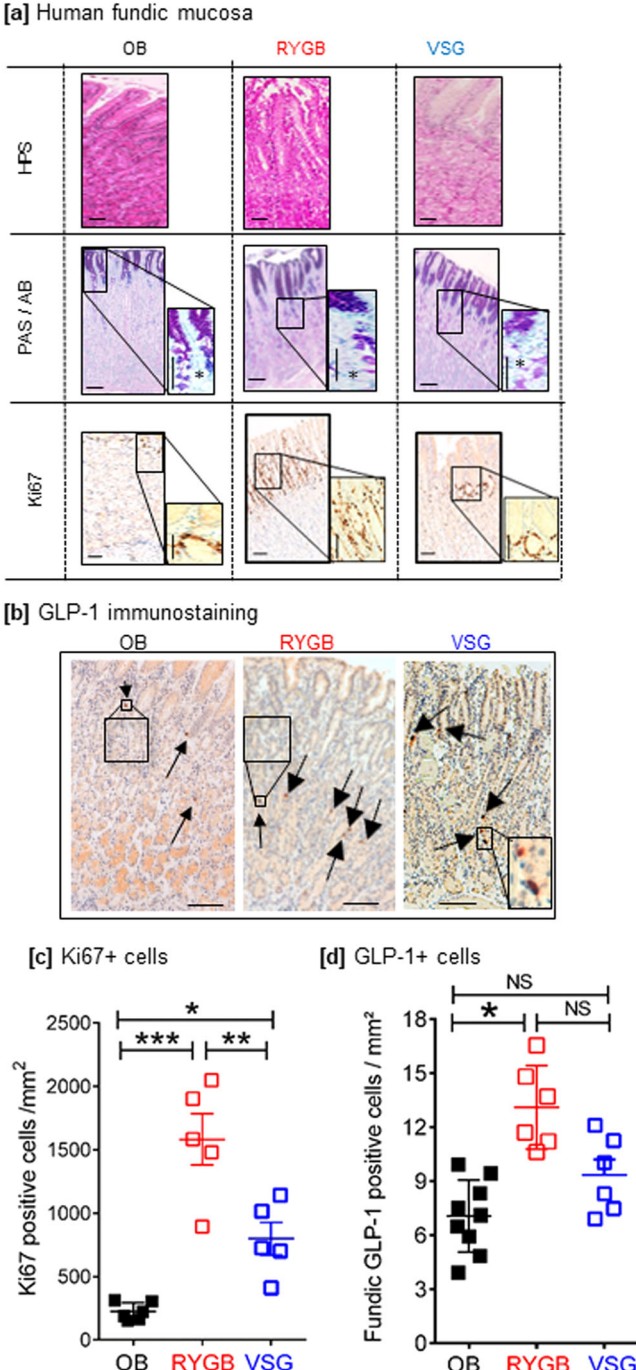

[a] Human fundic mucosa

[b] GLP-1 immunostaining

[c] Ki67+ cells

[d] GLP-1+ cells

**Fig. 3 Bariatric surgery induces remodeling of gastric mucosa in humans. a** Histological gastric mucosa phenotype before and after bariatric surgery in humans. Representative photomicrograph of HPS staining, Periodic Acid Schiff/Alcian blue (PAS/AB), and Ki67 immunostaining (brown nuclei) in fundic mucosa section from non-operated obese subjects (OB) vs. obese individuals that underwent RYGB or VSG surgery. Note, in VSG and RYGB vs. OB subjects, the increase in pit depth and hyperplasia of the surface mucous cells, the increase prominence of PAS-positive cells (that stain neutral mucin) in the neck region of the glands (asterisks in high magnification), and the higher Ki67-positive proliferative cells in the fundic mucosa. Bar scale 100 μm. **b** GLP-1 immunostaining in human fundic mucosa. Representative photomicrograph (arrows) of GLP-1-positive cells in fundic mucosa section from obese subjects vs. obese individuals who underwent RYGB and VSG surgery. Bar scale 100 μm. Quantification of Ki-67 proliferative cells (**c**) ($n = 6$ for OB, $n = 5$ for RYGB and VSG) and of GLP-1 immunoreactive cells (**d**) per mm² in the fundic mucosa of obese subjects (OB $n = 9$) and subjects operated on VSG ($n = 6$) or RYGB ($n = 6$). The data are presented as scatter data plots (each point represents one individual) with mean ± SEM and analyzed by one-way ANOVA followed by Tukey's multiple comparison test. *$P < 0.05$, **$P < 0.01$, and ***$P < 0.001$ and NS not significant. Source data are provided as a Source data file.

The released gastric active GLP-1 may not only stimulate local GLP-1 receptors resulting in reduced antro-duodenal contractility and delaying gastric emptying[31] but it may also induce activation of neural pathways located in the proximal gut to generate signals to the brain via the vagal afferent neurons as reported previously[32]. At this stage, we can state that the stomach contains functional GLP-1-producing cells that may contribute to local GLP-1 action and circulating GLP-1 levels and probably to the control of glucose homeostasis. Importantly, this transient rise of gastric GLP-1 secretion is abrogated in obese rats and an enhanced gastric emptying might amplify overeating and obesity[33,34].

Nowadays, the most effective therapy for morbid obesity is bariatric surgery since the procedure leads to sustained weight loss and resolves or improves obesity-associated type 2 diabetes[35–38]. This improvement of glucose homeostasis was suggested to involve, among numerous mechanisms, a strong increased in GLP-1 plasma levels in response to food intake. The VSG procedure consists of surgical removal of 80% of the corpus/fundus along the greater curvature leaving a narrow tube for transit of the ingested food. RYGB involves an extensive rearrangement of the upper gut, with the creation of a small gastric pouch, bypassing the remaining stomach and part of the small intestine by connecting the distal jejunum to the new gastric pouch. This procedure creates a "non-physiological situation" in which the ingested food directly enters the jejunum. Given the marked differences in post-surgical restructuration of the gastrointestinal tract by RYGB and VSG, the comparable clinical outcomes raise important questions about the involved mechanisms in their beneficial effects, e.g., increased levels of GLP-1. We report that the gastrointestinal remodeling by RYGB and VSG is associated with an increased density and the number of GLP-1-expressing cells in the gastric mucosa. In addition, we demonstrate in rats that the abrogation of gastric GLP-1 secretion in response to obesity is restored after VSG, indicating that these gastric GLP-1-expressing cells could have a relevance in response to bariatric surgery.

We propose that at least part of this outcome is related to changes in the number and sensitivity of gastric endocrine cells producing active GLP-1. However, despite being an attractive hypothesis, a definitive role for gastric GLP-1 in glucose homeostasis remains to be proven. Some non-exclusive possible mechanisms of action could be at play: stomach-produced GLP-1 could have a local rather than systemic action, as proposed for pancreatic GLP-1[7–9]. This local action of GLP-1 is consistent with activation of GLP-1 receptors on gastric epithelial cell receptors[10,11], the pylorus, and the vagal nerve terminals[13,15], which could result in a reduction in gastric emptying[39,40], one of the mechanisms of action of the drugs for diabetes therapy. Alternatively, transient release of gastric bioactive GLP-1 could directly enter the liver in quantities sufficient to affect hepatic glucose production and metabolism. This mechanism is supported by data reporting that intraportal GLP-1 stimulates hepatic vagal afferent activities[41] and synergistically increases glucose-stimulated insulin secretion[42,43], all effects that may contribute to maintenance of glucose homeostasis.

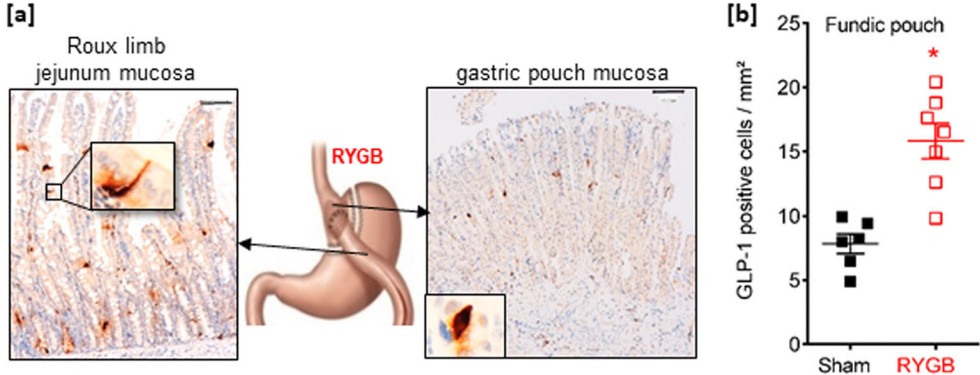

**Fig. 4 The density of fundic GLP-1-positive cells increases after RYGB surgery in rats. a** Representative photomicrograph showing few GLP-1-positive cells in the gastric pouch fundic mucosa and the alimentary Roux limb (RL) jejunum mucosa section from one RYGB-operated rat. Bar scale 100 μm. Insert: high magnification of a GLP-1-positive cell. The included image is the property of Johnson and Johnson and Ethicon Endo-Surgery (Europe) and is reproduced here with their kind permission. **b** Quantification of the number of GLP-1 immunoreactive cells per $mm^2$ in the fundic pouch. Data are presented as scatter data plots with mean ± SEM and were analyzed using Mann–Whitney test. *$P < 0.05$. Source data are provided as a Source data file.

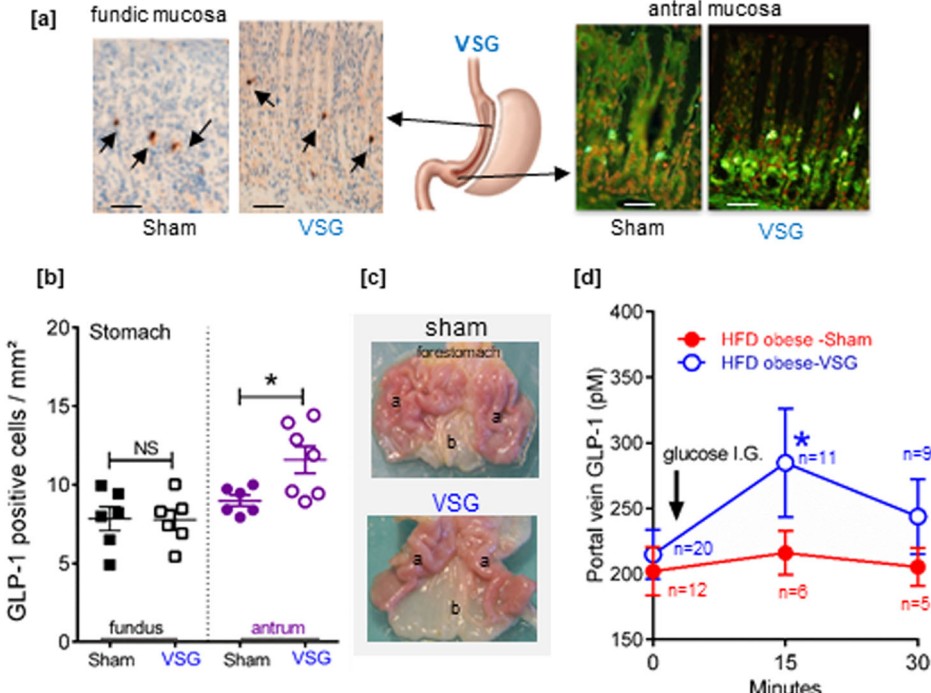

**Fig. 5 The density and functionality of GLP-1-positive cells in the antral mucosa increases after VSG surgery in rats. a** Representative photomicrographs of GLP-1 immunoreactive cells in the remaining fundic mucosa (left) and antral mucosa (right) in HFD obese rats operated on Sham or VSG. Bar scale 100 μm. Note that on right panel immunofluorescent GLP-1-positive cells are in green with nuclei in red. The included image is the property of Johnson and Johnson and Ethicon Endo-Surgery (Europe) and is reproduced here with their kind permission. **b** Quantification of the number of GLP-1 immunoreactive cells per $mm^2$ in the fundic and antral mucosa. The data are presented as scatter data plots (each point represents one rat) with mean ± SEM and analyzed using Mann–Whitney test *$P < 0.05$, NS not significant. **c** Postmortem macroscopic photomicrographs of stomach [fundus/corpus (a), antrum (b)] of HFD obese rats operated on sham or VSG, 2 weeks after surgery. Note the enlargement of the antrum (b) in VSG-rat. **d** Portal vein GLP-1 levels after an intragastric load of glucose (2 g/kg BW) in anesthetized sham- and VSG-operated HFD rats. Data represented the mean ± SEM of *n* values indicated at each time point on the graph for HFD obese Sham rats and HFD obese rats operated on VSG surgery. Data were analyzed by a two-way ANOVA. *$P < 0.05$ vs. sham. Source data are provided as a Source data file.

Further characterization of the stomach GLP-1-expressing cells reveals that these cells also contain ghrelin peptide, a finding consistent with PC1/3 requirement for processing pre-proglucagon into GLP-1[14] and pre-proghrelin into ghrelin[44]. Moreover, some of these gastric GLP-1-expressing cells also contain SST normally produced by endocrine D cells. These findings are compatible with the current view that endocrine cell populations exhibit more complex patterns of co-localization of

hormones than previously appreciated[45]. If ghrelin and SST are co-secreted with GLP-1, the biological actions of all these hormones have to be taken into account in the regulation of gastric functions.

Interestingly RYGB- and VSG-induced changes in gastric GLP-1-producing cells extend to other endocrine cells. Thus the increase in GLP-1-expressing cell density is associated with increased number of ghrelin-producing cells after RYGB, while in

VSG, it is associated with reduced density of ghrelin-, SST-, and 5HT-producing cells. The reduction in ghrelin cell number fits with our previously reported decrease in ghrelin mRNA levels in the remaining fundic mucosa of rats after VSG[16]. It is also consistent with studies in rat models of VSG and RYGB showing reduced plasma ghrelin levels after VSG but not after RYGB[46]. In fact, the reduced circulating ghrelin levels after RYGB remain controversial for humans in the literature. Systematic reviews and meta-analyses indicate that, in the majority of human studies[47], ghrelin levels decreased in the short term (≤3 months) but increased in the long term (>3 months). Accordingly, our current data show that RYGB individuals, operated on for >2 years, experienced increased number of ghrelin-producing cells in the gastric pouch. Collectively, these findings show that VSG and RYGB expand GLP-1-expressing cells in the rat and human stomach and affect the other endocrine X/A, D, and enterochromaffin cells. The precise mechanisms involved in this surgical procedure-specific "gastroplasticity" warrant further investigation. However, it is possible that these adaptive processes involve reprogramming of endocrine cell differentiation. The renewal of the gastric epithelium is ensured by stem cells located within the tubular glands[48], where a subpopulation of fundic chief cells, expressing Lgr5, has been identified and can function as injury-responsive "reserve" stem cells[49]. As the surgical restructuring of the gastrointestinal tract represents a mucosal assault that needs to be repaired, one can speculatively suggest that gastric epithelial cell renewal would be engaged into a new endocrine cell differentiation program. This is closed to data showing that HFD enhances intestinal stemness[50] and that calorie restriction can affect stem cell activity by augmenting the function of Paneth cells, which indirectly increases the number of intestinal crypt base columnar stem cells[50].

In summary, this report brings to light the presence of gastric GLP-1-positive cells whose density in the mucosa increases after RYGB and VSG surgeries. It highlights a potential role of stomach-derived GLP-1 acting locally and ascribes a role in physiology and metabolism. A better understanding of these adaptive mechanisms of gastric mucosa after RYGB and VSG and their exploitation will help in the development of new approaches to bariatric surgeries and possibly allow to replace them with less invasive and gastric-targeted endocrine therapies for the treatment of obesity-related diabetes.

## Methods

**Animal surgery and post-surgery procedures**. All animal studies were conducted in compliance with EU directives for animal experimentation and were approved by the Ethical Committee of Paris North and The French Minister of Higher Education, Research and Innovation (MESRI APAFIS #8290). Male Wistar rats weighting 220–240 g were fed an HFD (Altromin C45, Genestil, Royaucourt, France) for 4 months. Some of these rats were sacrificed and stomachs were used for ex vivo studies and the other HFD obese rats were randomly assigned to the RYGB, VSG, and sham-operated (sham) groups.

The first step of the surgeries is the removal of the nonglandular part of the stomach (forestomach) by application of an ETS-Flex 35-mm staple line (Ethicon, Issy les Moulineaux, France). For VSG procedure, 80% of gastric fundus was resected by application of a second ETS-Flex 35-mm staple line, leaving a thin gastric tube in continuity with the esophagus. For RYGB procedure, a second staple line (Proximate TX30V 30 mm, Ethicon), parallel to the first one, was applied to delimit a gastric pouch that represented 20% of the initial stomach size while preserving the arterial and venous supply. The jejunum was transected 20 cm after the pylorus. The alimentary limb was anastomosed to the gastric pouch, and the biliopancreatic limb was anastomosed 15 cm distally to gastrojejunal anastomosis. Sutures were made by a 7-0 Prolene wire (Ethicon). For sham-operated rats, the stomach was pinched with an unarmed staple gun and the jejunum was incised and stitched immediately after. All surgeries ended with 4-0 vicryl and 3-0 vicryl (Ethicon) sutures to sew up the abdominal wall and the skin, respectively.

Immediate post-operative care included daily evaluation of health and behavior of each animal, subcutaneous injections of 10 ml Bionolyte G5 (Baxter, Maurepas, France) once a day within 3 days. A free access to solid normal diet (Altromin 1324, Genestil, Royaucourt, France) was allowed from day 3. Sham-operated rats

had the same post-operative care as the surgical groups. After day 3, post-operative analyses consisted of daily monitoring of body weight (BW) and food intake. Whole-body composition was measured in unanesthetized rats before and after surgery using an EchoMRI 100 whole-body composition analyzer (EchoMRI, Houston, TX, USA). Two weeks after surgery, an oral glucose tolerance test after 16 h fasting was performed as previously described[16,17]. Finally, rats were food deprived overnight, euthanized, and gastrointestinal segments were sampled for analyses.

**Human gastric samples**. All patients gave a written informed consent and the study was approved by the Committee for the Protection of humans, Aulnay-sous-Bois, Seine Saint-Denis, France. For gastric biopsies, subjects with obesity and candidates for bariatric surgeries were recruited at the Integrated Center for Medical and Surgical Care of Obesity of our hospital (HUPNVS). The studies complied with all relevant regulations regarding the use of human study participants and were conducted in accordance to the criteria set by the Declaration of Helsinki. The patients met the following criteria for obesity surgery: body mass index of at least 40 or 35 kg/m$^2$ with at least two comorbidities (hypertension, type 2 diabetes, dyslipidemia, or obstructive sleep apnea syndrome). Surgical candidates had no history of abdominal surgery or large hiatal hernia (>5 cm) and no previous history of inflammatory diseases of the gastrointestinal tract, no peptic ulcer, no esophageal or gastrointestinal motility disorder such as gastroparesis, no ongoing treatment with a weight loss medication or anticoagulants, or nonsteroidal anti-inflammatory drugs, a good general health status, and no evidence of psychiatric problems. Sixteen patients [12 females/4 males] of median age 42 years [31–52; 95% confidence interval (CI), $N = 16$], with median BMI of 43.85 [37.3–58.79; 95% CI, $N = 16$] were recruited. During endoscopic examination, fundus and antrum biopsies were collected and immediately frozen in nitrogen liquid and stored at −80 °C until further protein extraction for RIA of total GLP-1.

For histology and immunohistochemistry studies, 21 patients treated or not by bariatric surgery were retrospectively selected from the files of the Department of Pathology, Bichat Hospital, Paris, France. Informed consent was obtained from all patients (Supplementary Table 3: characteristics of each patient).

The age of the different groups was: obese control [6 females (F)/2 males (M)] median age 42 years [35–51, 95% CI], VSG group [4 F/2 M] median age 43.6 years [35–52, 95% CI], RYGB group [5 F/1 M] median age 45 years [30–56, 95% CI], and there was no significant difference between groups.

For the BMI (kg/m$^2$): obese control group: median BMI 49.8 [41–56; 95% CI, $N = 9$], VSG group median BMI 40.05 ([32.3–42.3, 95% CI, $N = 6$] $P < 0.01$ vs. obese control), and RYGB group median BMI 36.70 ([27.8–42.3, 95% CI, $N = 6$], $P < 0.001$ vs. obese control). None of the individuals had diabetes or took medication to control glycemia.

**Reverse transcription and quantitative real-time PCR (qPCR)**. Total RNA was extracted from frozen fundus and antrum mucosal scrapings from lean or HFD obese rats with TRIzol reagent (Invitrogen, Saint Aubin, France). One microgram from each sample was converted to cDNA using the Verso cDNA Synthesis Kit (Thermo Scientific, Waltham, MA, USA). Primers (Supplementary Table 2) were designed using the Roche assay design center or based on previous studies. Eurofins Genomics (Ebersberg, Germany) synthesized all primers. Real-time qPCR was performed using the Light Cycler 480 system (Roche Diagnostics, Indianapolis, IN, USA) according to the manufacturer's instructions under the following conditions: 15 min denaturation at 95 °C followed by 40 cycles of 10 s at 95 °C, 45 s at 60 °C, and 10 s at 72 °C. Melting curves were performed for each reaction, from 55 °C to 95 °C at 0.11 °C/s. Ct values of the gene of interest were normalized with two different reference genes (L19 and HPRT1), which were chosen after multiple comparisons with numerous reference genes.

**Mucosal extracts for quantification of GLP-1**. Frozen lean or HFD obese rat gastric mucosal scrapings and human fundic and antral biopsies were homogenized (TissueLyser II, Retsch France) with an ice-cold buffer containing 1.5% HCl 1 M and 75% ethanol. Homogenates were centrifuged at 10,000 × g for 20 min at 4 °C, and supernatants were collected and subjected to hydrophobic chromatography (C18 SepPak cartridges, Waters Corporation) according to the manufacturer's instructions. Elution of the columns was performed with 1 ml of 80% (vol/vol) isopropanol/0.1% trifluoroacetic acid and dried completely in a vacuum concentrator.

For comparison purpose, the entire stomach (fundus+antrum), ileum, and colon mucosa were scrapped and homogenized as above in the presence of the DPP-4 inhibitor for total and active GLP-1 assay. For the total and active GLP-1 assay, dried pellets were resuspended in RIA buffer, and for MS analysis, dried pellets were resuspended in PBS, and in each case, the protein concentration was determined using the Pierce BCA Protein Assay Kit (Prod# 23225, ThermoFisher Scientific, France).

**RIA of total and active GLP-1**. RIA of total and active GLP-1 was performed using the Millipore RIA Kit Cat no. GLP1T-36HK and Cat no. GLP1A-35HK (ImmunoDiagnostics, France), respectively. GLP1T-36HK allows quantification of all forms of GLP-1 (7–36) amide, GLP-1 (7–37), GLP-1 (9–36) amide, GLP-1 (9–37),

GLP-1 (1–36) amide, and GLP-1 (1–37). The antibody used in the assay binds specifically to the C-terminal portion of GLP-1 and to both amidated and non-amidated forms, and GLP-1 (7–36) amide was used as standard and to prepare $^{125}$I radiolabeled GLP-1. The specificity of the assay was 100% GLP-1 [7–36], GLP-1 [9–36], and GLP-1 [7–37]. We further verified that there was no cross-reactivity with GLP-2 and glucagon (Supplementary Fig. 2). GLP1A-35HK allows quantification of the biologically active form of GLP-1 [i.e., GLP-1 (7–36) amide or GLP-1 (7–37)].

**LC-MS/MS acquisition.** Thirty micrograms of protein extracted in duplicates from the colon, ileum, and stomach were precipitated with acetone at −20 °C. Proteins were digested overnight at 37 °C by 3 sequencing grade proteases: trypsin (12.5 µg/ml; Promega, Madison, WI, USA) in 20 µl of 25 mM $NH_4HCO_3$, GluC protease (25 µg/ml; Promega, Madison, WI, USA) in 20 µl of 20 mM of phosphate buffer, and chymotrypsin (25 µg/ml; Sigma-Aldrich) in 20 µl of 100 mM Tris-HCl and 10 mM $CaCl_2$ pH 7.8. Peptides were desalted using ZipTip µ-C18 Pipette Tips (Thermo Scientific) and were analyzed by an Orbitrap Fusion Tribrid coupled to a Nano-LC Proxeon 1000 equipped with an easy spray ion source (all from Thermo Scientific). Peptides were separated by chromatography with the following parameters: Acclaim PepMap100 C18 pre-column (2 cm, 75 µm i.d., 3 µm, 100 Å), Pepmap-RSLC Proxeon C18 column (75 cm, 75 µm i.d., 2 µm, 100 Å), 300 nl/min flow rate, using a gradient rising from 95% solvent A (water, 0.1% formic acid) to 40% B (80% acetonitrile, 0.1% formic acid) in 120 min, followed by a column regeneration of 20 min, for a total run of 140 min. Peptides were analyzed in the Orbitrap cell, in full ion scan mode, at a resolution of 120,000 (at $m/z$ 200), with a mass range of $m/z$ 350–1550 and an AGC target of $2 \times 10^5$. Fragments were obtained by high collision-induced dissociation activation with a collisional energy of 30% and a quadrupole isolation window of 1.6 Da. MS/MS data were acquired in the Orbitrap cell in the top-speed mode, with a total cycle of 3 s at a resolution of 30,000, with an AGC target of $1 \times 10^4$, a dynamic exclusion of 50 s and an exclusion duration of 60 s. Precursor priority was highest charge state, followed by most intense. Peptides with charge states from 2 to 8 were selected for MS/MS acquisition. The maximum ion accumulation times were set to 250 ms for MS acquisition and 30 ms for MS/MS acquisition in parallelization mode. All MS and MS/MS data for protein samples were processed with the Proteome Discoverer software (Thermo Scientific, version 2.1) coupled to an in-house Mascot search engine (Matrix Science, version 2.5.1). The mass tolerance was set to 7 ppm for precursor ions and 0.5 Da for fragments. The enzyme parameter was set to semi-specific for each enzyme used (trypsin, Glu-C, Lys-C and chymotrypsin) to identify putative sites of endogenous proteases. The following variable modifications were allowed: oxidation (Met), phosphorylation (Ser, Thr), and acetylation (Protein N-term). The SwissProt database (02/2017) with the *Rattus norvegicus* taxonomy was used for the MS/MS identification step. Peptide identifications were validated using a 1% false discovery rate threshold calculated with the Percolator algorithm. Abundance of peptides and proteins were determined by the Precursor ion area detector node from the Proteome Discoverer 2.1 software.

**GLP-1 release from stomach and colon fragments incubated ex vivo.** Stomachs and colons of ND lean or HFD obese rats were collected after euthanasia, washed to remove the debris and then cut into small pieces. The small pieces of each segment were incubated in a 37 °C bath with stirring in medium containing PBS continuously gassed with 95% $O_2$–5% $CO_2$. After a 30-min stabilization incubation, PBS (without IBMX) and 100 µM 3-isobutyl-1-methylxanthine (with IBMX) (Sigma Chemicals, St Louis Mo) was added and incubated for 60 min so that each rat served as its own control. Afterwards, the supernatants were collected and the fragments snapped frozen prior to homogenization and protein quantification with the Pierce BCA Protein Assay Kit (Prod #23225, ThermoFisher Scientific, France). The supernatants were stored at −20 °C until GLP-1 RIA. The GLP-1 release (pM) in each incubated fragment was normalized to the quantity of protein determined with the Pierce BCA Protein Assay Kit (Prod# 23225, ThermoFisher Scientific, France).

**Histology, immunohistochemistry, and immunofluorescence studies.** Rat tissue samples were immediately fixed overnight in formalin for 24 h, dehydrated, and embedded in paraffin. Three-micron section of fundic mucosa were cut from each selected paraffin block. Hematoxylin Phloxine Saffron and Alcian Blue+PAS stainings were performed. Immunohistochemistry was carried out using an automated immunohistochemical stainer according to the manufacturer's guidelines (Bond-Max autostainer, Leica, Wetzlar, Germany), after dewaxing and rehydrating paraffin sections and antigen retrieval by pretreatment with high temperature at pH 9. Tissue sections were then immunostained with primary antibodies: monoclonal mouse anti-Human Ki67 (M7240; Dako) diluted 1:100, pH 9; monoclonal mouse GLP-1 antibody [8G9] (Abcam, Cambridge, UK MA) diluted 1:3000, pH 9, polyclonal Rabbit 5HT (serotonin) antibody (Immunostar WI, USA) diluted 1/10,000, monoclonal mouse/human SST ghrelin antibody (R&D Systems #MAB8200) or anti-SST antibody (Dako, #A0566) both used at a dilution 1:500. Substitution of the primary antibody with PBS was used as a negative control. Subsequently, tissues were incubated with a secondary antibody polymer for 10 min (Bond Polymer Refine detection; DS9800; Leica Microsystems) and developed

with DAB-Chromogranin. Three pathologists, A.S., A.C., and M.H., blinded to the clinical data evaluated the immunostainings. Each slide was scanned with an Aperio ScanScope CS System (Leica Microsystemes SAS, Nanterre, France). Morphometric analyses were performed using the Calopix Software version 4.1.0.4 (powdered by TRIBVN, Chatillon, France) on at least 2 sections per sample. The number of GLP-1 cells was evaluated in 3–4 cross-sections.

For immunofluorescence, the primary antibodies used were monoclonal [8G9] to GLP-1, monoclonal SST antibody (#MAB2358), and monoclonal Human/Mouse Ghrelin Antibody (R&D systems #MAB8200), and the secondary antibodies were Alexa Fluor® 488 AffiniPure Donkey Anti-Rat IgG and Cyanine Cy™3 AffiniPure Donkey Anti-Mouse IgG from Jackson ImmunoResearch (Europe Ltd.) and were used at a final dilution of 1/300. The nuclei were stained with propidium iodide from (Invitrogen France) or with Hoechst solution (Sigma Chemicals, St Louis, MO). The sections were observed with a fluorescence microscopy and analyzed by confocal microscopy Sp8 Leica (Leica microsystems, France)

**Measure of GLP-1 levels in the portal vein or gastric vein after stomach or duodenal load of glucose.** Wistar rats (ND lean, HFD obese, or HFD obese operated on VSG) were fasted overnight with free access to water. They were anesthetized with intramuscular injection of urethane (1.125 g/kg) and a catheter was introduced either into the portal vein or into the gastric vein. Thirty-five minutes (stabilization period) after completion of the surgery, the pylorus was clamped, and the rats received intragastric or intraduodenal load of glucose (2 g/kg BW) or PBS in a volume that did not distend the stomach. Blood samples were collected from the portal vein or the gastric vein before and 15 and 30 min after the load.

In another set of experiments, we measured total and active GLP-1 levels in the gastric vein after stomach load of glucose. To this end, blood samples were collected into heparinized tubes containing DPP-4 inhibitor (Roche Diagnostics, Indianapolis, IN, USA) from the gastric vein before and 15 and 30 min after the gastric load of glucose. Blood was centrifuged ($2500 \times g$, 4 °C, 10 min) and the plasmas were collected, immediately frozen, and stored at −20 °C until RIA for total and/or active GLP-1.

**Statistical analyses.** Data are expressed as scattered plots or means +/− standard error (SE) or standard error of the mean (SEM). One-way analysis of variance (ANOVA) followed by Tukey's multiple comparison test were used to compare more than two groups; nonparametric Mann–Whitney tests were used to compare two groups. A two-way ANOVA was used to compare time-dependent effect.

**Reporting summary.** Further information on research design is available in the Nature Research Reporting Summary linked to this article.

## Data availability
The complete data sets are available in the PRIDE partner repository under the identification number: PXD009867, http://www.ebi.ac.uk/pride/archive/projects/PXD009867. All other data supporting the findings of this study are available within the paper and its supplementary information files. Source data are provided with this paper.

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

## Acknowledgements

L.R.-P. was supported by FRM FDM20150633742; A.-C.J. and J.-B.C. by The French Minister of Higher Education and Research (MRT). This work was supported by INSERM, Université de Paris and by a Fare Grant from the French National Society of Gastroenterology (SNFGE) to A.B. M.L.G. received funding from Société Francophone du Diabète (SFD). The authors thank Olivier Thibaudeau and Laure Wingertsmann from the Histology Platform, Dounia Mansour from the department of Pathology for morphometric and quantitative analyses of the immunostainings, and Doris Da Silva from the Department of Digestive Surgery for helping with gastric vein experiments in vivo. The authors thank Professor Patricia Brubaker, Departments of Physiology and Medicine, Canada Research Chair in Vascular and Metabolic Biology, University of Toronto for critical reading of the manuscript.

## Author contributions

L.R.-P., A.-C.J., J.L.B., M.L.G., and A.B. conceived and designed the study and interpreted the data. L.R.-P., A.-C.J., J.-B.C., A.W., and A.B. performed the animal experiments and histological analyses of the rat tissues. A.W., A.B., and S.B. performed confocal analyses of the rat tissues. T.L. and M.B. performed the mass spectrometric analyses. L.R.-P., S.L., J.-P.M., and S.M. recruited the obese subjects and candidates for bariatric surgeries and A.-L.P. provided gastric biopsies. A.S., A.C., and M.H. performed the immunostainings and morphometric analyses of human and rat tissues. A.B. wrote the paper with help from J.L.B., S.L., Y.A., and M.L.G. All authors approved the final version of the manuscript.

## Competing interests

The authors declare no competing interests.
