## [Peer Review File · Nature Communications]

This manuscript has been previously reviewed at another journal that is not operating a transparent peer review scheme. This document only contains reviewer comments and rebuttal letters for versions considered at *Nature Communications*.

Reviewers' Comments:

Reviewer #1:

Remarks to the Author:

The authors have made improvements to the manuscript by addressing several issues raised in the initial review of the manuscript. Unfortunately, even though their findings illustrating an increase in GLP-1+ cells following bariatric surgery are novel, the main limitation of this study is that it remains largely descriptive with no clear mechanism behind these observations, nor the physiological relevance of these morphological changes following bariatric surgery. There are numerous troubling inconsistencies throughout the paper and the presentation of the data remains frequently suboptimal. I have simply listed some of the problems in point form followed by a more detailed set of comments in regards to experiments and figures.

Despite previous requests, there are no controls for small or large bowel shown in Figure 1A,B- hence the actual values presented, normalized to L19 are quantitatively meaningless to the reader. The authors frequently confuse measurement of RNA and protein with cell localization throughout the paper. For example, they state that the endocrine cells in the stomach express the Gcg and Pyy and Pcsk1 genes and refer to Figure 1, yet Figure 1 shows no cellular localization data whatsoever.

Why is the peptide content of GLP-1 2-fold higher in the fundus vs the antrum (Figure 1), yet throughout the rest of the paper, the peptide content is the same or greater in the antrum?

How should the average reader interpret the vast difference in quantitative assessment of Gcg peptides presented by the authors using quantitative proteomics, namely 15.9 to 155.9 times more abundance of these peptides in the ileum vs, the stomach- is this not extremely troubling, given that this data is described as "quantitative"?

In Fig 2A the difference between PBS and IBMX seems largely driven by 1 outlier. Remove this one data point and the majority of the values obtained for GLP-1 are the same. It is difficult to accept that as stated, this is 2-fold difference in GLP-1 secretion. Why is there no IBMX data shown for the antrum in the same Figure?

In Figure 2B, it is impossible to see the actual data since it is all "normalized". Moreover, the data presented shows that the peak GLP-1 response is identical whether or not the stimulus comes from the stomach or the duodenum. This makes the major point that counter the authors arguments- the distal gut likely secretes GLP-1 when nutrients are infused proximally via the well established proximal-distal axis of GLP-1 secretion. See Brubaker Holst and many others. Furthermore, the portal vein integrates all of the gut regions draining the GI tract and is not stomach-selective- the splenic vein might be more selective for the stomach. So there is no clear conclusion that can be drawn as to whether the stomach is actually contributing meaningfully to the GLP-1 release in any of these experiments.

Why does GLP-1+ cells go up in the antrum but not fundus after VSG (Fig 3) yet this switches completely in high fat fed animals and now the increase is observed in the fundus, but not the antrum in Figure 4- is this not troubling to the authors, notably since there is no explanation? Why do the authors use the term plasticity to refer to the changes in cell numbers described in Fig 2C? What do they mean?

What cells are proliferating in the data shown in Extended Data 4- would it not seem critical to identify the cells? This massive increase in proliferation would seem troubling for predisposing to cancer etc which has not been reported- is the same increase seen in the stomach of the animals after VSG or RYGB?

Why is there no plasma GLP-1 data that corresponds to the animals subjected to VSG or RYGB?

We are shown glucagon and leptin and insulin data (Extended Data 5) but no GLP-1, which seems to be the major point yet data is missing

Specific comments

1) It is unfortunate that the authors have decided against further characterizing these GLP-1+ cells in regards to whether these GLP-1+ cells colocalize with other endocrine cell markers, endocrine cell differentiation transcription factors, or PC1/3 (or PC2). In addition, it is difficult to

get a sense of how rare these GLP-1 immunoreactive cells are without comparison to other endocrine cell types in the gastric mucosa.

Is there a particular reason why a similar characterization of mRNA transcripts including *Gcg*, *Pcsk1*, *Pax6*, etc along the entire intestinal tract comparing NC to HFF rats, shown in the responses to reviewers' comments, was not included in the revised manuscript? It is somewhat confusing that some panels in Figure 1 compare NC to HFF rat, while others only show NC (Extended Data 1 A and B) or exclusively HFF rats (Extended Fig 1 C-H).

2) If the authors choose not to present data on other EEC lineages following bariatric surgery, some of their previous key findings regarding gastric remodeling following VSG and RYGB surgeries in rats (ie expansion of MNCs, and changes in other EEC mRNA transcripts) should at least be mentioned in the manuscript for clarification.

3) The data in Figure 2A is now indicates that it was normalized to protein content of the fragments. However the graph appears exactly the same as the panel shown in the original manuscript that did not indicate normalization. I trust this was simply an error in y-axis labelling in the original manuscript?

The authors should clearly explain how this experiment was performed in the methods section as it was described in the Responses to Reviewers' Comments for clarity.

4) The same concerns regarding the interpretation of the data following intragastric glucose loading in Fig 2B still remain. As several studies have implicated that neural or paracrine mechanisms in the proximal small bowel stimulate GLP-1 secretion from L cells in the distal gut, and both the stomach and intestine drain into the portal vein, the possibility of neural transmission (for example) following intragastric glucose loading to more distal L cells in the intestine was not explored. It is also unexpected that portal vein GLP-1 levels are similar 15 min following a gastric and duodenal load of glucose. Do portal vein GLP-1 levels differ at earlier time points? How does portal vein GLP-1 levels compare to intragastric glucose loading without pylorus clamping? Do gastric ex vivo cultures secrete GLP-1 in response to glucose? How does this compare to GLP-1 secretion from duodenum ex vivo cultures following glucose exposure?

5) The authors state on line 170 that "increased mucosa GLP-1 expressing cells occurs exclusively in the stomach". This statement is misleading. Although the density of GLP-1+ cells did not increase in the ileum or colon following RYGB in this study, there was significant intestinal expansion in the ileum and colon implying there are more GLP-1+ cells, thus this finding is not exclusive to the stomach.

6) As the total GLP-1 assay used in this study detects all forms of GLP-1, is the GLP-1 released from these gastric cultures ex vivo biologically active?

7) It is somewhat paradoxical that *Gcg* and *Pcsk1/3* mRNA levels are similar, albeit variable, between fundus and antrum under both NC and HFF rats (Fig 1A,B), yet mucosal GLP-1 content and basal GLP-1 secretion are ~2 fold higher in fundus than in the antrum (Fig 1D and Fig 2A). In addition, the density of GLP-1+ cells in sham operated HFF rats are similar between fundus and antrum (Fig 3A), whereas this was not quantified under NC conditions. Again it would be beneficial to show mRNA levels, GLP-1+ cell density, mucosal content etc in one figure under the same conditions as it is difficult to make meaningful conclusions the way the data is currently presented.

8) What is the physiological importance of this finding (ie ↑GLP-1+ cells following gastric bypass surgery)? The authors hypothesize that part of the improvement in glucose homeostasis following bariatric surgery in part could be explained by changes in GLP-1 production by gastric cells. This was primarily based on the observation that GLP-1+ cells were higher in the fundic pouch of rats and in biopsies of human patients following RYGB surgery, and in the antral mucosal of rats following VSG.

Although GLP-1+ cells may increase in the remnant stomach compared to the same area of sham operated rats, the majority of the fundus and atrum (containing GLP-1+ cells) are excluded following surgery, particularly after RYGB. Consequently it is difficult to conceptualize the authors'

theory based on these observational studies alone. Does the remnant stomach (eg residual fundus) secrete more GLP-1 than the fundus of sham operated rats, or respond to lower concentrations of glucose or other nutrients?

Minor points

1) 2 panels are labeled F in Extended Data 1

2) Why was immunofluorescence rather than immunohistochemistry of GLP-1+ cells performed in the antral mucosa in Fig 3? It is difficult to visualize positive cells as there appears to be a high degree of background staining or autofluorescence.

3) According to Dey et al, *Endocrinology* 146 (2):713-727 (2005), there are some discrepancies of regarding PC1/3, PC2 cleavage sites shown in Fig 1G.

4) Some figures are shown as individual data points while others are shown as whiskers min to max. It would be beneficial to the reader to present one or the other (preferably scatter plots).

5) In Figure 2: panel A and B present rat data, whereas panel C and D present human data following bariatric surgery. For ease of reading perhaps the human data could be presented together with the remaining human data, and the rat data (2A and B) moved to Fig 1?

6) In legend of Figure 1: this figure legend could be reduced by excluding some sections that should be in the methods section. For example "Tissue sections were formalin-fixed....counterstained with Mayer's haemalum", and "The extracts were subjected.....dried in a vacuum concentrator". "Data Availability....." is already mentioned in the methods section. In legend of Figure 2, Page 20, "The formalin-fixed tissues embedded in paraffin blocks were retrospectively selected cut into 3µm sections. The slides were immunolabelled with GLP-1 antibody [8G9] diluted 1:3,000. Nuclei were counterstained with Mayer'shaemalum." should be move to material and methods.

7) Human primers for GCG, PCSK2 and PCSK1/3 are listed in Extended Table 3, yet no human qPCR is shown in the manuscript

8) Stats are not shown Fig1 D and F

9) Extended Table 1 should include an explanation of what the columns represent (eg mascot score, endogenous protease signature) and the definition of what the values in these columns implies (eg MH+) such that the reader can more readily interpret the data.

Reviewer #2:

Remarks to the Author:

The revised manuscript has addressed most of this reviewer's concerns, although more thorough characterization of the GLP1+ cells is preferred. Observation of this new cell lineage in the stomach is the major novelty of this manuscript. Since not too much functional data are provided, the authors should at least provide detailed molecular and cellular characterization data of the potentially interesting cell type.

Responses to Reviewers' comments:

Reviewers' comments:

Reviewer #1 (Remarks to the Author):

The authors have made improvements to the manuscript by addressing several issues raised in the initial review of the manuscript. Unfortunately, even though their findings illustrating an increase in GLP-1+ cells following bariatric surgery are novel, the main limitation of this study is that it remains largely descriptive with no clear mechanism behind these observations, nor the physiological relevance of these morphological changes following bariatric surgery.

We thank the reviewer for acknowledging the novelty of our study, we have now acquired additional data to better characterize the gastric GLP-1 positive cells. Moreover, we propose a functional impact of the increased number of these cells after bariatric surgery, as they are associated with a restoration of the early peak of GLP-1 in response to a gastric load of glucose that was abrogated in response to obesity.

Thanks to the comments of both reviewers, we think that the robustness of the data quality of the manuscript have been improved.

There are numerous troubling inconsistencies throughout the paper and the presentation of the data remains frequently suboptimal. I have simply listed some of the problems in point form followed by a more detailed set of comments in regards to experiments and figures. Despite previous requests, there are no controls for small or large bowel shown in Figure 1A,B -hence the actual values presented, normalized to L19 are quantitatively meaningless to the reader.

We have now introduced the controls in the manuscript as Supplementary data 1 showing *Gcg*, *Pcsk1/3*, *Pax6* and *Pyy* mRNA levels along the gastrointestinal mucosa from the stomach to the colon of rat fed ND or HFD during 4 months.

The authors frequently confuse measurement of RNA and protein with cell localization throughout the paper. For example, they state that the endocrine cells in the stomach express the *Gcg* and *Pyy* and *Pcsk1* genes and refer to Figure 1, yet Figure 1 shows no cellular localization data whatsoever.

The manuscript has now been carefully edited to avoid confusion between RNA, protein and cell localization.

Why is the peptide content of GLP-1 2-fold higher in the fundus vs the antrum (Figure 1), yet throughout the rest of the paper, the peptide content is the same or greater in the antrum?

One need to distinguish the quantity of GLP-1 in the tissue (*New Figure 1A*) from the quantity of GLP-1 secreted by the tissue at the basal levels or upon stimulation (*New Figure 2*). It is possible that some cells are producing GLP-1 in the fundus but these cells are not secreting as much GLP-1 in the antrum explaining the apparent discrepancies.

Using a different in vitro method we have performed additional sets of experiments with stomach and colon explants (*cf Material & Methods page 20, results page 5 2nd paragraph*). The data (*new Figure 2*) show similar basal release of total GLP-1 in rat corpus/fundus and antrum, and a 15-fold higher total GLP-1 release from the rat colon whereas the content of GLP-1 is only 5 times higher in the colon than in the fundus and 10-times higher in the colon than in the antrum (Figure 1A).

How should the average reader interpret the vast difference in quantitative assessment of *Gcg* peptides presented by the authors using quantitative proteomics, namely 15.9 to 155.9 times more

abundance of these peptides in the ileum vs, the stomach-is this not extremely troubling, given that this data is described as “quantitative”?

The ileum/gastric ratio is indeed different according to the enzyme used for the analyses. This is due to the capacity of the MS to identify specific peptides that are different depending of the enzyme. One can only compare ratio of expression from a single enzyme treatment (i.e. ileum/gastric versus colon/gastric after trypsin digestion) but one cannot compare two ratios obtained from two different enzymes (ileum/gastric after trypsin versus ileum/gastric after GluC). The main notion to be retained from these quantifications is that the ileum contains much more GLP-1 than the stomach.

In Fig 2A the difference between PBS and IBMX seems largely driven by 1 outlier. Remove this one data point and the majority of the values obtained for GLP-1 are the same. It is difficult to accept that as stated, this is 2-fold difference in GLP-1 secretion. Why is there no IBMX data shown for the antrum in the same Figure?

We have performed additional measurements of PBS and IBMX release of GLP1 from fundus, antrum and colon explants. As shown in new Fig 2 A, GLP-1 release upon IBMX stimulation is similar in the 3 tissues.

In Figure 2B, it is impossible to see the actual data since it is all “normalized”. Moreover, the data presented shows that the peak GLP-1 response is identical whether or not the stimulus comes from the stomach or the duodenum. This makes the major point that counter the authors arguments-the distal gut likely secretes GLP-1 when nutrients are infused proximally via the well established proximal-distal axis of GLP-1 secretion. See Brubaker Holst and many others. Furthermore, the portal vein integrates all of the gut regions draining the GI tract and is not stomach-selective-the splenic vein might be more selective for the stomach. So there is no clear conclusion that can be drawn as to whether the stomach is actually contributing meaningfully to the GLP-1 release in any of these experiments.

We respectfully disagree with this comment, the origin of GLP-1 secretion in response to nutrients is still a matter of debate, in particular as nutrient cannot directly stimulate distal L cells as soon as 15min after ingestion. We do not deny the “proximal-distal axis of GLP-1 expression (and not secretion)”, but we propose that in addition to the secretion of GLP-1 by distal L cells, a gastric source of GLP-1 can contribute to the early peak of GLP-1 in the portal vein.

We agree that “the portal vein integrates all of the gut regions draining the GI tract and is not stomach-selective” this is exactly why we directly compare the rise of GLP-1 in this vein and not in the splenic vein.

We expressed the data as % of basal, to be able to pull different sets of experiments acquired over time.

Why does GLP-1+ cells go up in the antrum but not fundus after VSG (Fig 3) yet this switches completely in high fat fed animals and now the increases is observed in the fundus, but not the antrum in Figure 4-is this not troubling to the authors, notably since there is no explanation? What do they mean? As we already suggested, GLP-1 positive cells may be different cells in the antrum and in the fundus, their GLP-1 secretion capacity seems to differ and their different adaptive response HFD or to VSG confirm this hypothesis.

In the previous version of the manuscript, Figure 3 referred to modification after VSG whereas Figure 4 referred to modification after RYGB. As we have already published, jejunal adaptation and modification

of GLP-1 positive cell number and density were completely different after those 2 surgeries (Cavin et al. 2016 Gastroenterology). We also have already described different rat gastric adaptation but not related to GLP-1 (Arapis et al Plos One 2015)

To illustrate these differences, we have now extended our analyses to human gastric adaptation after RYGB and VSG (new Figure 3) and extend to the ileum and colon, the adaptation in rats after RYGB (new Figure 4, and supplementary figure 6) and VSG (new Figure 5, and supplementary figure 6).

It is remarkable that GLP-1 positive cell density are modified similarly in the fundus of human and rat, in both GLP-1 positive cell density is specifically increased after RYGB but not after VSG (compare Figure 3 with Figures 4 and 5).

Thus, the data in obese rats and individuals are concordant and indicate the implication of distinct mechanisms specific to each surgical procedure.

Why do the authors use the term plasticity to refer to the changes in cell numbers described in Fig 2C?

Plasticity of the cells can be defined as the capacity of these cells to modify their number and/or their function in response to external or internal cues/signals (see Le Gall et al. Nutr Rev. 2019;77(3):129-143). In this study, plasticity includes change in the number of enteroendocrine cells

What cells are proliferating in the data shown in Extended Data 4-would it not seem critical to identify the cells? This massive increase in proliferation would seem troubling for predisposing to cancer etc which has not been reported-is the same increase seen in the stomach of the animals after VSG or RYGB?

The previously extended data 4 is now presented as new Figure 3 as we realized from the reviewer's comment that this set of experiment was interesting for the readers.

The proliferating cells are presently the mucus-secreting columnar cells and some mucous neck cells, the proliferating region of the fundic part of the stomach. We previously reported a hyperplasia of the jejunum after RYGB with an increased proliferating rate of the jejunal crypts in both rats and humans (Cavin Gastroenterology 2016). Similarly we show here an increased proliferation of the fundus in human confirming what we previously reported in rats (Arapis et al PLoSOne 2015).

Still, we agree with the reviewer, no gastric or small bowel cancer has been reported after RYGB, except one case report but the cancer was in the excluded part of the stomach. This observation suggests that this massive hyperplasia is still physiologic and does not correspond to a neoplasia.

Why is there no plasma GLP-1 data that corresponds to the animals subjected to VSG or RYGB? We are shown glucagon and leptin and insulin data (Extended Data 5) but no GLP-1, which seems to be the major point yet data is missing.

We apologize for this omission that has now been corrected and the data are presented in new supplementary Figure 5 Panel G.

Specific comments

1) It is unfortunate that the authors have decided against further characterizing these GLP-1+ cells in regards to whether these GLP-1+ cells colocalize with other endocrine cell markers, endocrine cell differentiation transcription factors, or PC1/3 (or PC2). In addition, it is difficult to get a sense of how rare these GLP-1 immunoreactive cells are without comparison to other endocrine cell types in the

gastric

mucosa.

We have now further characterized the gastric GLP-1 positive cell population and found that some of these cells contain ghrelin, a finding consistent with PC1/3 requirement for processing pre-proglucagon into GLP-1 (*Drucker DJ et al. J Clin Invest. 127: 4217; 2017*) and pre-proghrelin into ghrelin. These data revealed the scarcity of GLP-1 + cells with approximately one GLP-1 + cell for 10 ghrelin + cells in human fundic mucosa (see Figure 3D vs Supplementary Data 4).

Moreover, some of these GLP-1 positive cells also contain somatostatin produced by endocrine D cells. These findings are compatible with the current view that endocrine cell populations exhibit more complex patterns of co-localization of hormones than previously appreciated (*Habib AM et al. Endocrinology 153:3054-; 2012*).

Is there a particular reason why a similar characterization of mRNA transcripts including Gcg, Pcsk1, Pax6, etc along the entire intestinal tract comparing NC to HFF rats, shown in the responses to reviewers' comments, was not included in the revised manuscript? It is somewhat confusing that some panels in Figure 1 compare NC to HFF rat, while others only show NC (Extended Data 1 A and B) or exclusively HFF rats (Extended Fig 1 C-H).

This point has now been taken and the results are presented in new Supplementary Figure 1.

2) If the authors choose not to present data on other EEC lineages following bariatric surgery, some of their previous key findings regarding gastric remodeling following VSG and RYGB surgeries in rats (ie expansion of MNCs, and changes in other EEC mRNA transcripts) should at least be mentioned in the manuscript for clarification.

We have now introduced in the manuscript new data extending to other gastric endocrine cells the RYGB- and VSG-induced changes we previously reported only in gastric GLP-1 producing cells (supplementary data 4) .

After RYGB, the increase in GLP-1 positive cell density is associated with an increased density of ghrelin-producing cells, while after VSG, we report a reduced density of ghrelin-producing cells, of somatostatin- D cells and of 5HT-producing EC cells but no modification of GLP-1 positive cell density. These observations in human fundic mucosa are complementary of the data we already reported in rats for ghrelin (*Arapis PLoS One 2015*)

This surgical procedure-specific "gastroplasticity" of endocrine cells argues for distinct mechanisms in the differentiation program of gastric endocrine cell lineage. The underlying mechanism have now to be deciphered but we think it is a beyond the scope of this study.

3) The data in Figure 2A is now indicates that it was normalized to protein content of the fragments. However, the graph appears the same as the panel shown in the original manuscript that did not indicate normalization. I trust this was simply an error in y-axis labelling in the original manuscript? The authors should clearly explain how this experiment was performed in the methods section as it was described in the Responses to Reviewers' Comments for clarity.

This has now been clearly stated in material & Methods page 21

4) The same concerns regarding the interpretation of the data following intragastric glucose loading in Fig 2B still remain. As several studies have implicated that neural or paracrine mechanisms in the proximal small bowel stimulate GLP-1 secretion from L cells in the distal gut, and both the stomach and

intestine drain into the portal vein, the possibility of neural transmission (for example) following intragastric glucose loading to more distal L cells in the intestine was not explored.

We try to address that specific point by using vagotomy rats but due to technical problems the results were not conclusive

It is also unexpected that portal vein GLP-1 levels are similar 15 min following a gastric and duodenal load of glucose. Do portal vein GLP-1 levels differ at earlier time points?

Due to the difficulty of these physiological experiments, we have not performed earlier time points.

How does portal vein GLP-1 levels compare to intragastric glucose loading without pylorus clamping?

Do gastric ex vivo cultures secrete GLP-1 in response to glucose?

How does this compare to GLP-1 secretion from duodenum ex vivo cultures following glucose exposure?

We did not measure total GLP-1 secretion in gastric or duodenal *ex-vivo* cultures following glucose exposure. We did not address these specific points as they were not required in the previous run of questions from the reviewer

5) The authors state on line 170 that “increased mucosa GLP-1 expressing cells occurs exclusively in the stomach”. This statement is misleading. Although the density of GLP-1+ cells did not increase in the ileum or colon following RYGB in this study, there was significant intestinal expansion in the ileum and colon implying there are more GLP-1+ cells, thus this finding is not exclusive to the stomach. We agree with the reviewer that previously there was some confusion between cell density and cell number that has now been corrected in the revised manuscript.

6) As the total GLP-1 assay used in this study detects all forms of GLP-1, is the GLP-1 released from these gastric cultures ex vivo biologically active?

As stated in the material and method, we measured total GLP-1. We cannot answer on the activity of GLP-1 released from gastric *ex-vivo* cultures. This is an interesting point that was not previously addressed and that would require more additional experiments

7) It is somewhat paradoxical that Gcg and Pcsk1/3 mRNA levels are similar, albeit variable, between fundus and antrum under both NC and HFF rats (Fig 1A,B), yet mucosal GLP-1 content and basal GLP-1 secretion are ~2 fold higher in fundus than in the antrum (Fig 1D and Fig 2A). In addition, the density of GLP-1+ cells in sham operated HFF rats are similar between fundus and antrum (Fig 3A), whereas this was not quantified under NC conditions. Again it would be beneficial to show mRNA levels, GLP-1+ cell density, mucosal content etc in one figure under the same conditions as it is difficult to make meaningful conclusions the way the data is currently presented.

Once again, one need to distinguish the number of GLP-1 expressing cells from the quantity of GLP-1 produced by each cell (and thus the tissue) from the quantity of GLP-1 secreted by the tissue at the basal levels or upon stimulation. It is possible that some cells are producing GLP-1 in the fundus but these cells are not secreting as much GLP1 as the cells in the antrum explaining the apparent discrepancies. It has been previously demonstrated that the sensitivity of GLP-1 expressing cells (in the distal intestine) could be different in response to metabolic changes despite similar levels of GLP-1 mRNA or protein or cell number contributing to the confusion (see Desaulcy et al. Endocrinology 2016).

We hope the new organization of the data will make it more clear to the readers

8) What is the physiological importance of this finding (ie ↑GLP-1+ cells following gastric bypass surgery)? The authors hypothesize that part of the improvement in glucose homeostasis following bariatric surgery in part could be explained by changes in GLP-1 production by gastric cells. This was primarily based on the observation that GLP-1+ cells were higher in the fundic pouch of rats and in biopsies of human patients following RYGB surgery, and in the antral mucosal of rats following VSG. Although GLP-1+ cells may increase in the remnant stomach compared to the same area of sham operated rats, the majority of the fundus and atrum (containing GLP-1+ cells) are excluded following surgery, particularly after RYGB. Consequently it is difficult to conceptualize the authors' theory based on these observational studies alone. Does the remnant stomach (eg residual fundus) secrete more GLP-1 than the fundus of sham operated rats, or respond to lower concentrations of glucose or other nutrients?

We now propose (*Discussion section, page 9, lines 6-18*) that, the beneficial outcomes after RYGB and VSG, are at least partly related to changes in the number of gastric endocrine cells producing GLP-1 contributing to the restoration of early GLP-1 secretion after a gastric load of glucose (new Figure 5). However, despite being an attractive hypothesis, a definitive role for gastric GLP-1 in glucose homeostasis remains to be proven. Some possible mechanisms of action can be at play: first, stomach-produced GLP-1 could have a local rather than systemic action, through activation of GLP-1 receptors on epithelial cells, the pylorus and the vagal nerve terminals leading to reduction in gastric emptying, one of the mechanisms of action of GLP-1. Alternatively, gastric GLP-1 release could enter the liver in quantities sufficient to affect hepatic glucose production and metabolism. This is consistent with data showing that intraportal GLP-1 stimulates hepatic vagal afferent activities (*Nakabayashi H, Am J Physiol. 271: E808-, 1996*) and, synergistically increases glucose-stimulated insulin secretion (*Ionut V Diabetologia 48: 967-, 2005; Balkan B, Li X Am. J. Physiol. 279: R1449-, 2000*), all effects that may contribute to maintenance of glucose homeostasis.

Minor points

1) 2 panels are labeled F in Extended Data 1

This has been corrected as the manuscript and associated figures have now been completely modified and reorganized.

2) Why was immunofluorescence rather than immunohistochemistry of GLP-1+ cells performed in the antral mucosa in Fig 3? It is difficult to visualize positive cells as there appears to be a high degree of background staining or autofluorescence.

To be honest we began these studies with immunofluorescence of GLP-1+ in the antral mucosa and shift thereafter for quantitative purpose to IHC.

3) According to Dey et al, *Endocrinology* 146 (2):713-727 (2005), there are some discrepancies of regarding PC1/3, PC2 cleavage sites shown in Fig 1G.

The cleavage sites of PC1/3 and PC2 were corrected in the new Figure 1B.

4) Some figures are shown as individual data points while others are shown as shown as whiskers

min to max. It would be beneficial to the reader to present one or the other (preferably scatter plots).

Newer expectations for more informative publication figures encourage the use of scatter-dot plots to showcase individual data points, thus we now present the main figures as individual data points with mean and SEM.

5) In Figure 2: panel A and B present rat data, whereas panel C and D present human data following bariatric surgery. For ease of reading perhaps the human data could be presented together with the remaining human data, and the rat data (2A and B) moved to Fig 1?

This has been corrected as the manuscript and associated figures have now been completely modified and reorganized.

6) In legend of Figure 1: this figure legend could be reduced by excluding some sections that should be in the methods section. For example "Tissue sections were formalin-fixed....counterstained with Mayer's haemalum", and "The extracts were subjected....dried in a vacuum concentrator". "Data Availability....." is already mentioned in the methods section.

The legend of the new Fig1 was rewritten accordingly to your suggestion.

In legend of Figure 2, Page 20, "The formalin-fixed tissues embedded in paraffin blocks were retrospectively selected cut into 3µm sections. The slides were immunolabelled with GLP-1 antibody [8G9] diluted 1:3,000. Nuclei were counterstained with Mayer'shaemalum." should be move to material and methods.

This has been corrected

7) Human primers for GCG, PCSK2 and PCSK1/3 are listed in Extended Table 3, yet no human qPCR is shown in the manuscript

This has been corrected

8) Stats are not shown Fig1 D and F

This has been corrected

9) Extended Table 1 should include an explanation of what the columns represent (eg mascot score, endogenous protease signature) and the definition of what the values in these columns implies (eg MH+) such that the reader can more readily interpret the data.

We now provided a legend to the Supplementary Table 1

Reviewer #2 (Remarks to the Author):

The revised manuscript has addressed most of this reviewer's concerns, although more thorough characterization of the GLP-1+ cells is preferred. Observation of this new cell lineage in the stomach is the major novelty of this manuscript. Since not too much functional data are provided, the authors should at least provide detailed molecular and cellular characterization data of the potentially interesting cell type.

As previously described, we have now acquired additional data to better characterize the gastric GLP-1 expressing cells. Moreover, we propose a functional impact of the increased number of these cells after bariatric surgery, as they are associated with a restoration of the early peak of GLP-1 in response to a gastric load of glucose that was abrogated in response to obesity

Thanks to the comments of both reviewers, we think that the robustness of the data quality of the manuscript have been improved.

Reviewers' Comments:

Reviewer #1:

Remarks to the Author:

The authors have again made improvements to the manuscript and have addressed the majority of concerns raised over the course of the reviewing process. Despite the novelty of their findings regarding an increase in GLP-1+ immunopositive cells following bariatric surgery, and the morphological changes including gastric endocrine cells in the human fundic mucosa in gastric bypass subjects, the study remains primarily descriptive with little functional data. There is no compelling data presented to suggest that the putative GLP-1+ gastric GLP-1 cells are actually functionally important

It is appealing that the increase in antral GLP-1+ cells following VSG correlates with a restoration of the abrogated GLP-1 peak in the portal vein following intragastric glucose loading of HFD obese rats, however the same concerns previously raised regarding the interpretation of this intragastric loading/portal vein data remain. Furthermore, if this indeed reflects gastric-derived GLP-1 secretion, then this data suggests that gastric GLP-1 secretion is longer glucose responsive (either directly or indirectly) under HFD conditions with no other data to complement this finding.

- It is also troubling that fasted portal GLP-1 levels are similar between sham and VSG HFD rats (Fig 5 F) yet fasted plasma GLP-1 levels in VSG HFD rats is ~ 2 fold higher than sham HFD rats (sup Data 5). Would portal vein GLP-1 levels not also be much higher in fasted VSG HFD rats than sham HFD rats?

While it is appreciated that each ex vivo culture has now been used as its own control as requested, it would have been beneficial to also include fragments that were not treated with IBMX as a control for basal GLP-1 secretion over the subsequent 60 min incubation period that was used for IBMX

- Why are there fewer points for IBMX vs T0 if each sample was its own control?

Other points:

- 1) Figure 1 is mislabeled according to main text and figure legend
- 2) In Figure 1 A: GLP-1+ cells are labelled GLP-1 IR (bottom of each IHC panel) does the IR mean immunoreactivity? Should be defined in legend, or the IR should be taken out as readers may misinterpret the labeling as staining for the GLP-1R (GLP-1 receptor)
- 3) The legend for Figure 5 is mislabeled, and the main text does not correspond to the correct panels in Figure 5
- 4) Quality of main figures remains very poor (pixilated) which is troubling
- 5) In contrast to GLP-1+ immunostaining in the human fundic mucosa described here, the recent study by Roberts et al., 2019 Diabetes did not detect proglucagon derived peptides in their mass spec analysis of human gastric mucosa. This disparity should at least be commented on in the Discussion
- 6) It is mentioned in the text that some of the GLP-1+ cells colocalize with ghrelin or SST (Fig 1 C), however it appears that both of these markers colocalize with every GLP-1+ cell. It is difficult to get a sense of what "some" means without quantification or different representative photomicrographs.
- 7) While normalized or percent basal data can be presented, this does not mean the actual real data in their original units should not be presented in the main Figures

Reviewer #2:

Remarks to the Author:

The authors have shown additional characterization data. However, the findings of GLP1+ cells in the stomach is not new (upon reading more literature). There are a number of reports showing the presence of the cells in the tissue. To the credit of the authors they show the increased numbers of

GLP+ cells are increased in the various part of the stomach upon bariatric surgeries. That said, the study lacks of mechanistic insights.

Responses to Reviewers' comments

Reviewer #1 (Remarks to the Author):

Reviewer's comment. *The authors have again made improvements to the manuscript and have addressed the majority of concerns raised over the course of the reviewing process. Despite the novelty of their findings regarding an increase in GLP-1+ immunopositive cells following bariatric surgery, and the morphological changes including gastric endocrine cells in the human fundic mucosa in gastric bypass subjects, the study remains primarily descriptive with little functional data. There is no compelling data presented to suggest that the putative GLP-1+ gastric GLP-1 cells are actually functionally important*

It is appealing that the increase in antral GLP-1+ cells following VSG correlates with a restoration of the abrogated GLP-1 peak in the portal vein following intragastric glucose loading of HFD obese rats, however the same concerns previously raised regarding the interpretation of this intragastric loading/portal vein data remain. Furthermore, if this indeed reflects gastric-derived GLP-1 secretion, than this data suggests that gastric GLP-1 secretion is longer glucose responsive (either directly or indirectly) under HFD conditions with no other data to complement this finding.

(NB: all the modifications are in red in the revised manuscript)

Authors's answer. To address these specific points, we have conducted challenging new experiments in which GLP-1 levels were determined in the gastric vein upon a gastric load of glucose. As shown in the revised **Figure 2B**, the basal portal vein GLP-1 levels are significantly 2.7-fold higher ($P < 0.01$) than the levels in the gastric vein. An intragastric load of glucose (*in a volume that did not induce any distension of the stomach*) led to a significant and rapid (t 15 min) increase of GLP-1 levels in both the portal vein (+45%, $P < 0.05$ vs. PBS) and the gastric vein (+60%, $P < 0.05$ vs PBS 15 min) arguing for the contribution of the gastric-derived GLP-1 secretion to the portal vein GLP-1 levels. These data are discussed **lines 224-229**.

To complement this finding and to address the responsiveness of the gastric GLP-1 positive cells upon stimulation, additional **ex-vivo culture experiments, using fundus and antral fragments from HFD obese rat** were done. As shown in the new **supplementary Data 4** and **lines 136-140** of the revised MS), IBMX still stimulates GLP-1 release in the fundus of obese rats but, the stimulation in the antrum of obese rats was reduced. Although we do not have a clear explanation for these opposite effects, their result would be a reduction in the total secretion of gastric GLP-1. Thus, the stomach contains functional cells able to produce and secrete GLP-1 upon stimulation, an effect that is impaired in the stomach of obese rat.

Reviewer's comment - *It is also troubling that fasted portal GLP-1 levels are similar between sham and VSG HFD rats (Fig 5 F) yet fasted plasma GLP-1 levels in VSG HFD rats is ~ 2 fold higher than sham HFD rats (sup Data 5). Would portal vein GLP-1 levels not also be much higher in fasted VSG HFD rats than sham HFD rats?*

Authors's answer. We understand the reviewer's trouble, however one cannot compare the fasted plasma GLP-1 levels in VSG HFD and sham rats (present Supplementary Data 6 G) in blood collected from the tail of fasted awake animals, with GLP-1 levels in blood collected from the portal vein in fasted urethane-anesthetized VSG HFD and sham rats (Figure 5F).

Other points:

- 1) Figure 1 is mislabeled according to main text and figure legend. This was corrected in the revised version
- 2) In Figure 1 A: GLP-1+ cells are labelled GLP-1 IR (bottom of each IHC panel) does the IR mean immunoreactivity? Should be defined in legend, or the IR should be taken out as readers may misinterpret the labeling as staining for the GLP-1R (GLP-1 receptor).

This was taken into account and now modified in the revised Figure 1A (as GLP-1 immunostaining) in order to avoid confusion

- 3) The legend for Figure 5 is mislabeled, and the main text does not correspond to the correct panels in Figure 5

This was now corrected in the revised version

- 4) Quality of main figures remains very poor (pixilated) which is troubling In contrast to GLP-1+ immunostaining in the human fundic mucosa described here, the recent study by Roberts et al., 2019 Diabetes did not detect proglucagon derived peptides in their mass spec analysis of human gastric mucosa. This disparity should at least be commented on in the Discussion

The quality of the main figures has been improved. We now discuss the recent studies by Roberts et al., 2019 (see lines 213-216 of the revised MS).

- 5) It is mentioned in the text that some of the GLP-1+ cells colocalize with ghrelin or SST (Fig 1 C), however it appears that both of these markers colocalize with every GLP-1+ cell. It is difficult to get a sense of what “some” means without quantification or different representative photomicrographs.

We agree with the reviewer and we deleted “some” in the sentence

- 6) While normalized or percent basal data can be presented, this does not mean the actual real data in their original units should not be presented in the main Figures

In the revised manuscript, we now included the real data (See revised Fig 2A left) and Figure 2B

The authors again thank the reviewer 1 for acknowledging the novelty of our study, we have now performed additional experiments (that were a challenge for us) and the data show the contribution of the gastric-derived GLP-1 secretion to the portal vein GLP-1 levels.

Reviewer #2 (Remarks to the Author):

The authors have shown additional characterization data. However, the findings of GLP1+ cells in the stomach is not new (upon reading more literature). There are a number of reports showing the presence of the cells in the tissue. To the credit of the authors, they show the increased numbers of GLP1+ cells are increased in the various part of the stomach upon bariatric surgeries. That said, the study lacks of mechanistic insights.

We respectfully disagree with this view of reviewer 2. Even if reviewer 2 is already convinced of the presence of GLP-1 in the stomach, reserves of reviewer 1 suggest this fact is not obvious for all the community and need to be strengthened by more experimental data as the ones reported by our study. In addition, we are now showing modifications of GLP-1 cell number and functionality after bariatric surgery. That said, we agree with reviewer that the underlying mechanisms have now to be deciphered. Undoubtedly, this will be the scope of future studies.

Reviewers' Comments:

Reviewer #3:

Remarks to the Author:

The authors have adequately addressed most of the concerns of previous reviewer 1. However:

a) It is odd that in the new data measuring gastric and portal GLP-1 in the response to a gastric glucose load, the increment in GLP-1 concentration in the portal vein is higher than that in the gastric vein. If all the GLP-1 released in response to gastric glucose instillation arose from a gastric GLP-1 source, the GLP-1 increment should be lower in the portal vein, since the gastric vein blood is diluted by other portal blood at this point. It is still therefore possible that the GLP-1 measured in this experiment arises elsewhere. It would have been very helpful if the authors had measured peripheral venous blood in parallel in this experiment, so we could identify how much of the pre-stimulus GLP-1 arises from the stomach, rather than reflecting the concentration in arterial blood supplying the stomach.

b) Reviewer 1 previously commented that it was surprising that in peripheral blood, fasting GLP-1 is higher in RYGB or VSG compared with control, whereas in portal blood, fasting GLP-1 was similar in VSG vs sham. The reviewer was not comparing peripheral with portal blood, which have understandably been collected under different conditions. Responding that you cannot compare peripheral with portal blood is not an adequate response. I agree, it is surprising, as originally picked up by this reviewer.

However, I have some additional concerns:

1. The authors should include reference to older literature from the Unger group that the stomach is known to produce glucagon (rather than GLP-1). Since the stomach is known to produce glucagon, it is likely also to produce inactive GLP-1 fragments, as also found in pancreatic alpha cells which produce longer inactive GLP-1 forms starting at the 1-position.
2. The GLP-1 RIA uses an antibody directed against the C-terminus of GLP-1, and cannot distinguish the sequence at the N-terminus. The authors cannot therefore conclude whether the stomach may be producing inactive GLP-1(1-36/7) rather than active GLP-1(7-36/7) forms.
3. The GLP-1 antibody used for immunostaining, is similarly directed against the C-terminus of GLP-1, and cannot show that the stomach is producing active GLP-1(7-36/7) rather than inactive GLP-1(1-36/7) forms.
4. The mass spectrometry shows that the stomach produces proglucagon peptide, but not that it produces active GLP-1, since the identified peptides could originate from enzymatic digestion of proglucagon (which is known to be expressed – see above), rather than enzymatic digestion of active GLP-1.
5. The authors should consider quantifying the tissue content of active GLP-1 in stomach compared with more distal regions of the gut, to back up their claim that gastric active GLP-1 is a significant physiological player.
6. In several places the authors have used the units pM per weight of tissue. This is a meaningless number, since the concentration (pM) is dependent on the volume used for the assay. The values should be expressed in pmoles per unit weight.

Overall, it is essential that the authors show that they are measuring active GLP-1 from the stomach, rather than inactive GLP-1(1-36/7) peptides that are a by-product of proglucagon processing and do not contribute to functional circulating GLP-1 activity as they are not activated in the circulation. None of the experiments in the current manuscript can distinguish active GLP-1 from longer inactive GLP-1 fragments from proglucagon. This could be done using an active GLP-1 immunoassay. An example experiment could be: showing that active GLP-1 is produced in the gastric vein, and that gastric vein concentrations of active GLP-1 are higher than in venous blood. It would also be necessary to show that active GLP-1 production by the stomach is enhanced by bariatric procedures, to support the manuscript's claim that gastric GLP-1 is enhanced in these conditions.

Responses to Reviewer #3 (Remarks to the Author):

Reviewer 3's comment on our answers to reviewer 1's previous concerns

The authors have adequately addressed most of the concerns of previous reviewer

Authors:

We thank the reviewer for acknowledging how we have already improved our manuscript.

1. However:a) It is odd that in the new data measuring gastric and portal GLP-1 in the response to a gastric glucose load, the increment in GLP-1 concentration in the portal vein is higher than that in the gastric vein. If all the GLP-1 released in response to gastric glucose instillation arose from a gastric GLP-1 source, the GLP-1 increment should be lower in the portal vein, since the gastric vein blood is diluted by other portal blood at this point. It is still therefore possible that the GLP-1 measured in this experiment arises elsewhere.

Authors:

We respectfully disagree with the reviewer interpretation of our observations presented in Figure 2. Basal GLP-1 levels in the gastric vein and the portal vein were different and indeed portal GLP-1 is higher than gastric GLP-1. However, in response to a gastric glucose load, the increment in total GLP-1 concentration was +133pM in the portal vein (compared to a gastric PBS load at 15min) and +140pM in the gastric vein (compared to a gastric PBS load at 15 min). Both total GLP-1 increments were thus comparable and actually, they were not statistically different. Although we cannot formally exclude that some of the total GLP-1 measured in the portal vein arises elsewhere, the GLP-1 detected in the gastric vein was produced by the stomach and contributed to the increment of GLP-1 in the portal vein. *(These results are presented in the revised Figure 2 and revised manuscript lines 145-163).*

It would have been very helpful if the authors had measured peripheral venous blood in parallel in this experiment, so we could identify how much of the pre-stimulus GLP-1 arises from the stomach, rather than reflecting the concentration in arterial blood supplying the stomach.

Authors:

We tried very hard to set up the experiment suggested by the reviewer to measure GLP-1 in the peripheral venous blood and gastric vein at the same time but those experiments were impossible to be done with confidence and reproducibility and we could not find any example of this kind of rat experiment in the literature.

Reviewer's comment

b) Reviewer 1 previously commented that it was surprising that in peripheral blood, fasting GLP-1 is higher in RYGB or VSG compared with control, whereas in portal blood, fasting GLP-1 was similar in VSG vs sham. The reviewer was not comparing peripheral with portal blood, which have understandably been collected under different conditions. Responding that you cannot compare peripheral with portal blood is not an adequate response. I agree, it is surprising, as originally picked up by this reviewer.

Authors:

We apologize as we made a mistake in the previous data presented on Supplementary Data 6G. Actually, the previous data corresponded to GLP-1 concentration in response to a glucose

load and not to the fasted levels of GLP-1. To clarify this point we are now presenting the fasted and stimulated levels of Insulin and GLP-1 on this figure (*See Supplementary data 6 and supplemental Text 1*).

Reviewer 3's additional concerns

1. *The authors should include reference to older literature from the Unger group that the stomach is known to produce glucagon (rather than GLP-1). Since the stomach is known to produce glucagon, it is likely also to produce inactive GLP-1 fragments, as also found in pancreatic alpha cells which produce longer inactive GLP-1 forms starting at the 1-position.*

Authors:

We have now quoted 2 references from the Unger Group showing that the stomach produce glucagon [Munöz-Barragann *et al. Horm. Metab Res* 9(1) 37-39, 1977; Schusdziarra *et al. Am. J. Physiol.* 238(2) G109-113, 1980] (see revised manuscript lines 223-224)

2. *The GLP-1 RIA uses an antibody directed against the C-terminus of GLP-1, and GLP-1, and cannot distinguish the sequence at the N-terminus. The authors cannot therefore conclude whether the stomach may be producing inactive GLP-1(1-36/7) rather than active GLP-1(7-36/7) forms.*

3. *The GLP-1 antibody used for immunostaining, is similarly directed against the C-terminus of GLP-1, and cannot show that the stomach is producing active GLP-1(7-36/7) rather than inactive GLP-1(1-36/7) forms.*

4. *The mass spectrometry shows that the stomach produces proglucagon peptide, but not that it produces active GLP-1, since the identified peptides could originate from enzymatic digestion of proglucagon (which is known to be expressed – see above), rather than enzymatic digestion of active GLP-1.*

5. *The authors should consider quantifying the tissue content of active GLP-1 in stomach compared with more distal regions of the gut, to back up their claim that gastric active GLP-1 is a significant physiological player*

Authors:

We can only agree with this new point raised by reviewer #3, all our previous experiments could not formally distinguish inactive GLP-1(1-36/7) from active GLP-1(7-36/7). To address this concern, we have now performed additional experiments to quantify total and active GLP-1 content in the stomach and in distal regions of the gut i.e. the ileum and the colon (**new Figure 1 E&F**). Most importantly, we have measured total and active GLP-1 in the gastric vein in response to a gastric load of glucose (**Figure 2 E&F**).

Radioimmunoassays of total and active GLP-1 were performed using Millipore RIA kit Cat no. **GLP1T-36HK** that allows quantification of all forms of GLP-1 and, **GLP1A-35HK** that allows quantification of the biologically active form of GLP-1 [i.e. GLP-1 (7-36/7)].

As shown in revised Figure 1E& F, Total GLP-1 concentrations in the ileum and the colon are 3-fold and 6-fold higher respectively compared to total GLP-1 concentrations in the stomach (fundus & antrum together). However, active GLP-1 concentrations in the ileum and colon are actually only 2-fold and 1.5 fold higher respectively than in the stomach (Figure 3B). Thus, the stomach contains functional GLP-1 cells able to produce active GLP-1 and the putative contribution of the stomach in active GLP1 production may have been underestimated so far.

The new data presented in the Figure 2 show that in the gastric vein, the rise in total GLP-1 in response to a gastric glucose load was accompanied by a rise in active GLP-1 indicating that active GLP-1 is released from the stomach into the gastric vein and may contribute to the rise of GLP-1 reported in the portal vein.

Overall, it is essential that the authors show that they are measuring active GLP-1 from the stomach, rather than inactive GLP-1(1-36/7) peptides that are a by-product of proglucagon processing and do not contribute to functional circulating GLP-1 activity as they are not activated in the circulation. None of the experiments in the current manuscript can distinguish active GLP-1 from longer inactive GLP-1 fragments from proglucagon. This could be done using an active GLP-1 immunoassay. An example experiment could be: showing that active GLP-1 is produced in the gastric vein, and that gastric vein concentrations of active GLP-1 are higher than in venous blood. It would also be necessary to show that active GLP-1 production by the stomach is enhanced by bariatric procedures, to support the manuscript's claim that gastric GLP-1 is enhanced in these conditions.

Authors:

We thank the reviewer for insisting on this pertinent point and as stated before we have now measured active GLP-1 in different conditions either in total stomach extracts (Figure 1) and in the gastric vein in response to a gastric glucose load (Figure 2)

As previously explained the exact experiment suggested by the reviewer to measure active GLP-1 in the venous blood and the gastric vein simultaneously was impossible to be realized with enough confidence despite our numerous assays. It was thus inconceivable to be realized in animals after bariatric procedure.

Still we think that the new set of data we are presenting now, in complement with all the data already present in the previous version of the manuscript, show that the stomach contains functional GLP-1 positive cells able to produce and release active GLP-1 in the gastric vein which may contribute to the beneficial effects of bariatric procedures.

Reviewers' Comments:

Reviewer #3:

Remarks to the Author:

I thank the reviewers for having measured active as well as total GLP-1. I believe this has improved the manuscript

Answers to REVIEWERS' COMMENTS

Reviewer #3 (Remarks to the Author):

I thank the reviewers for having measured active as well as total GLP-1. I believe this has improved the manuscript

Authors : We would like to thank reviewer #3, for his/her previous challenging comments that as he/she recognized allow us to largely improved the quality of the manuscript